# Loss of *Grem1*-lineage chondrogenic progenitor cells causes osteoarthritis

Jia Q. Ng [1,10], Toghrul H. Jafarov[2,10], Christopher B. Little [3], Tongtong Wang[1,4], Abdullah M. Ali [2], Yan Ma[2], Georgette A. Radford[1], Laura Vrbanac[1], Mari Ichinose[1], Samuel Whittle[1,5], David J. Hunter [6], Tamsin R. M. Lannagan[1], Nobumi Suzuki[1], Jarrad M. Goyne[4], Hiroki Kobayashi [1], Timothy C. Wang[7], David R. Haynes[1], Danijela Menicanin[1], Stan Gronthos [1,8], Daniel L. Worthley [4,9,11] ✉, Susan L. Woods [1,4,11] ✉ & Siddhartha Mukherjee [2,11] ✉

Osteoarthritis (OA) is characterised by an irreversible degeneration of articular cartilage. Here we show that the BMP-antagonist *Gremlin 1* (*Grem1*) marks a bipotent chondrogenic and osteogenic progenitor cell population within the articular surface. Notably, these progenitors are depleted by injury-induced OA and increasing age. OA is also caused by ablation of *Grem1* cells in mice. Transcriptomic and functional analysis in mice found that articular surface *Grem1*-lineage cells are dependent on *Foxo1* and ablation of *Foxo1* in *Grem1*-lineage cells caused OA. FGFR3 signalling was confirmed as a promising therapeutic pathway by administration of pathway activator, FGF18, resulting in *Grem1*-lineage chondrocyte progenitor cell proliferation, increased cartilage thickness and reduced OA. These findings suggest that OA, in part, is caused by mechanical, developmental or age-related attrition of *Grem1* expressing articular cartilage progenitor cells. These cells, and the FGFR3 signalling pathway that sustains them, may be effective future targets for biological management of OA.

OA causes a considerable global health burden, affecting 10 percent of people over the age of 60[1]. OA affects all tissues of the joint including loss of AC, subchondral bone remodelling, osteophyte formation, synovial inflammation and joint capsule fibrosis, resulting in joint instability, pain and disability[2]. About 1 in 10 patients with knee OA will progress to joint replacement due to failure of medical care[3]. There is no cure for OA and existing treatments involve pain management and

lifestyle modification[4]. There are several mouse models that induce OA via the surgical destabilization of medial meniscus (DMM) or collagenase-induced injury to destabilize tendons and ligaments (collagenase-induced OA, or CIOA).

Articular cartilage (AC) is comprised of specialised chondrocytes that secrete a rich extracellular matrix (ECM) with high proteoglycan content, to allow effective movement between two bones[5,6]. Unlike

[1]Adelaide Medical School, Faculty of Health and Medical Sciences, University of Adelaide, Adelaide, SA, Australia. [2]Department of Medicine, Columbia University Medical Center, New York, NY, USA. [3]Raymond Purves Bone & Joint Research Laboratories, Kolling Institute, University of Sydney Faculty of Medicine and Health, Royal North Shore Hospital, St. Leonards, NSW, Australia. [4]Precision Cancer Medicine Theme, South Australian Health and Medical Research Institute, Adelaide, SA, Australia. [5]Rheumatology Unit, The Queen Elizabeth Hospital, Woodville South, SA, Australia. [6]Northern Clinical School, University of Sydney, St. Leonards, Sydney, NSW, Australia. [7]Department of Medicine and Irving Cancer Research Center, Columbia University, New York, NY, USA. [8]School of Biomedicine, Faculty of Health and Medical Sciences, University of Adelaide, Adelaide, SA, Australia. [9]Colonoscopy Clinic, Brisbane, QLD, Australia. [10]These authors contributed equally: Jia Q. Ng, Toghrul H. Jafarov. [11]These authors jointly supervised this work: Daniel L. Worthley, Susan L. Woods, Siddhartha Mukherjee. ✉e-mail: dan@colonoscopyclinic.com.au; susan.woods@adelaide.edu.au; sm3252@cumc.columbia.edu

growth plate (GP) cartilage, AC is a permanent tissue that requires the support of self-renewing progenitor cells to repopulate resident chondrocytes[5,7–11]. AC currently has limited regenerative capacity, with OA often resulting from injury, chronic mechanical stress or increasing age, with AC loss a key feature of OA[5,12,13].

Studies have confirmed skeletal stem cells[14] (SSC) based on immunophenotype[15] or lineage tracing[16] give rise to bone, cartilage and stroma but not fat lineages. Independently, a bone-fat (but not cartilage) progenitor cell population, marked by the expression of the *Leptin Receptor* (*Lepr*), was identified within the bone marrow[17]. The bone-cartilage-stromal progenitors in the growth plate and the bone-fat progenitors in the marrow express different markers (*Grem1* and *Lepr*, respectively) and have different roles during development and repair. Our initial study[16] focused on *Grem1*-lineage tissue-resident SSCs in the GP of mice. A subsequent study found that tissue-resident SSC can be activated to make AC using microfractures in conjunction with BMP and VEGF signalling; however, the location of these AC-forming SSCs remained unknown, and these stimuli appeared to generate only cartilage, not subchondral bone[13].

Here, we show that a population of *Grem1*-lineage chondrogenic progenitor (CP) cells, distinct from GP-resident SCCs, resides on the articular surface, and generates AC (and, in later stages, subchondral bone). We focus on the fate and function of these articular surface CP cells during aging, and upon OA-inducing injury using two independent models of OA (DMM and CIOA). We find that a *Grem1* expressing CP *Grem1*-lineage is lost through apoptosis in our OA models and that ablation of these CPs in young mice also causes OA. Single cell RNA sequencing (scRNAseq) of *Grem1*-expressing cells at the articular surface reveals distinct molecular features of *Grem1*-lineage cells and identifies FGFR3 signalling as a potential therapeutic target for *Grem1*-lineage CP cell maintenance and expansion. The FGFR3 ligand, FGF18 (the active agent in Sprifermin) is currently in human clinical trial for OA treatment[18]. Injection of FGF18 into injured joints increases the number of *Grem1*-lineage CP cells (but not mitosis in hypertrophic chondrocytes) and ameliorates OA pathology. Our study identifies a *Grem1*-expressing progenitor cell origin of articular cartilage, which is maintained, at least in part, by FGFR3 signalling, and when depleted results in OA.

## Results

### *Grem1* CP cells are depleted in OA

Focusing our studies on the knee joint of adult mice, we observed two anatomically distinct populations in the AC and GP marked by *Grem1*-lineage tracing (Fig. 1a). Physically, these were separated in the femur by 800–1900 microns.

OA is predisposed by injuries that destabilise the periarticular tissues of a joint[19]. We used two models of induced OA in mice, involving surgical DMM or collagenase VII degradation of intra-articular stabilising ligaments (CIOA), to examine the fate of *Grem1* lineage cells, *Lepr* bone marrow derived-mesenchymal stem cells (MSC) and articular chondrocytes marked by *Acan* in OA. *Rosa-TdTomato* reporter mice crossed with *Grem1-creERT*, *Acan-creERT* and *Lepr-cre* (a constitutive *Cre* line), henceforth termed *Grem1-TdT*, *Acan-TdT* and *Lepr-TdT* mice respectively. These mice faithfully labelled *Grem1-*, *Acan-* and *Lepr-* cells and, depending on the time point following *Cre* mediated recombination of the reporter, their respective progeny.

The inducible Cre lines, *Acan-TdT* and *Grem1-TdT*, were administered tamoxifen at 8 to 10 weeks of age, with DMM surgery (Fig. 1b, Supplementary Fig. 1a) performed 2 weeks later. *Lepr-TdT* mice of similar age had DMM surgery for comparison (Fig. 1b). DMM surgery results in a significant decrease in proliferating cells in both the superficial and non-calcified zones of the AC (Supplementary Fig. 1b). OA pathology was confirmed by loss of proteoglycans, surface fibrillation (Fig. 1c) and osteophyte formation (Fig. 1d). Quantification of the

total percentage of *Acan*, *Grem1* and *Lepr* traced AC cells at the site of proteoglycan loss showed a significant decrease in the *Grem1*-lineage population only (Fig. 1e, f). This suggested *Grem1* AC chondrocytes may be important in maintaining AC integrity and protection from daily mechanical insult. The absence of lineage-traced cells in the AC of *Lepr-TdT* mice is consistent with a prior study[17] and suggested that the primary role of the *Lepr*-lineage is in the haematopoietic niche in diaphyseal bone marrow[20,21]. Notably, the persistence of *Acan* AC chondrocytes in DMM animals suggests that the initial stage of OA is not due to total chondrocyte loss, but rather is specific to the loss of the *Grem1*-lineage CP population.

In the CIOA model (Fig. 1g), we observed more severe OA pathology as expected[22]. As with DMM, *Grem1-TdT* CIOA mice exhibited significant loss of *Grem1*-lineage CP cells through apoptosis, decreased AC thickness and increased OA pathology[23] compared to PBS injected animals (Fig. 1h–k, Supplementary Fig. 1c). This suggested *Grem1*-lineage articular surface CP cells may be important in the pathogenesis of OA.

### *Grem1* marks a bipotent chondrogenic and osteogenic progenitor population in the AC

Given that OA models were characterised by loss of *Grem1*-lineage CP cells, we tested whether *Grem1*-lineage CP cells were in fact resident stem-progenitor cells for normal articular cartilage post-natal development and maintenance. *Grem1-TdT* and *Acan-TdT* mice were administered tamoxifen at postnatal day 4 to 6 before sacrifice with age matched *Lepr-TdT* mice, to determine the contribution of each lineage to the AC (Fig. 2a). *Grem1*-lineage CP cells were immediately observed within the cartilaginous epiphysis and meniscus after a week, and had given rise to 39.4% of the AC. In later stages of development, we found the *Grem1*-lineage CP cells also generated osteoblasts in subchondral bone, and by 1 month had populated the entire joint structure, including 30.7% of the AC (Fig. 2b). A partially overlapping distribution of AC cells was observed with *Acan* progeny (Fig. 2b). In contrast, *Lepr*-lineage cells[24] were not evident within the AC but were found as peri-sinusoidal cells in the bone marrow, consistent with a previous report[20] (Fig. 2b). Immunofluorescence staining at 20 weeks (Fig. 2c and Supplementary Fig. 2a) showed that *Grem1* and *Acan* cells gave rise to SOX9+ and hypertrophic COLX+ chondrocytes, as well as OCN+ osteoblasts. In contrast, *Lepr* cells only gave rise to OCN+ osteoblasts, very occasional SOX9+ chondrocytes and no COLX+ chondrocytes (Fig. 2c, Supplementary Fig. 2a).

To further analyse the clonogenic and differentiation potential of *Grem1*-lineage CP cells in the early adult knee ex vivo, *Grem1-TdT* and *Acan-TdT* mice were administered tamoxifen at 6 weeks of age and knee joints harvested 2 weeks later (Fig. 2d). *Grem1* labelled specific articular surface cells (Fig. 2e) overlapping with, but more limited than, the total *Acan* chondrocyte population (Supplementary Fig. 2b). These *Grem1-* and *Acan*-traced cells from the AC of the tibiofemoral joint were isolated via flow cytometry and plated at clonal density (Fig. 2e). As *Lepr* cells were not found in the AC (Supplementary Fig. 2c), a *Lepr* comparable population could not be included in this AC-specific experiment. *Grem1*-lineage articular CP clones were more frequently capable of serial propagation, in contrast to *Acan* clones (Fig. 2f). Furthermore, *Grem1* expression was significantly higher in *Acan* clones capable of in vitro expansion compared to those that were not (Supplementary Fig. 2d), suggesting *Grem1* expression correlated with self-renewal in vitro. Of the *Grem1-* and *Acan*-lineage clones that underwent >3 passages, no significant difference in CFU-F efficiency was observed (Fig. 2g). When subjected to multilineage differentiation, both adult *Grem1-* and *Acan*-lineage clones gave rise to osteogenic (alizarin red stain+) and chondrogenic (alcian blue stain+), but not adipogenic (oil red O stain+) progeny (Fig. 2h, Supplementary Fig. 2e). This is consistent with previous studies on postnatal whole bone populations[16]. A significantly greater percentage of *Grem1*-lineage clones were capable

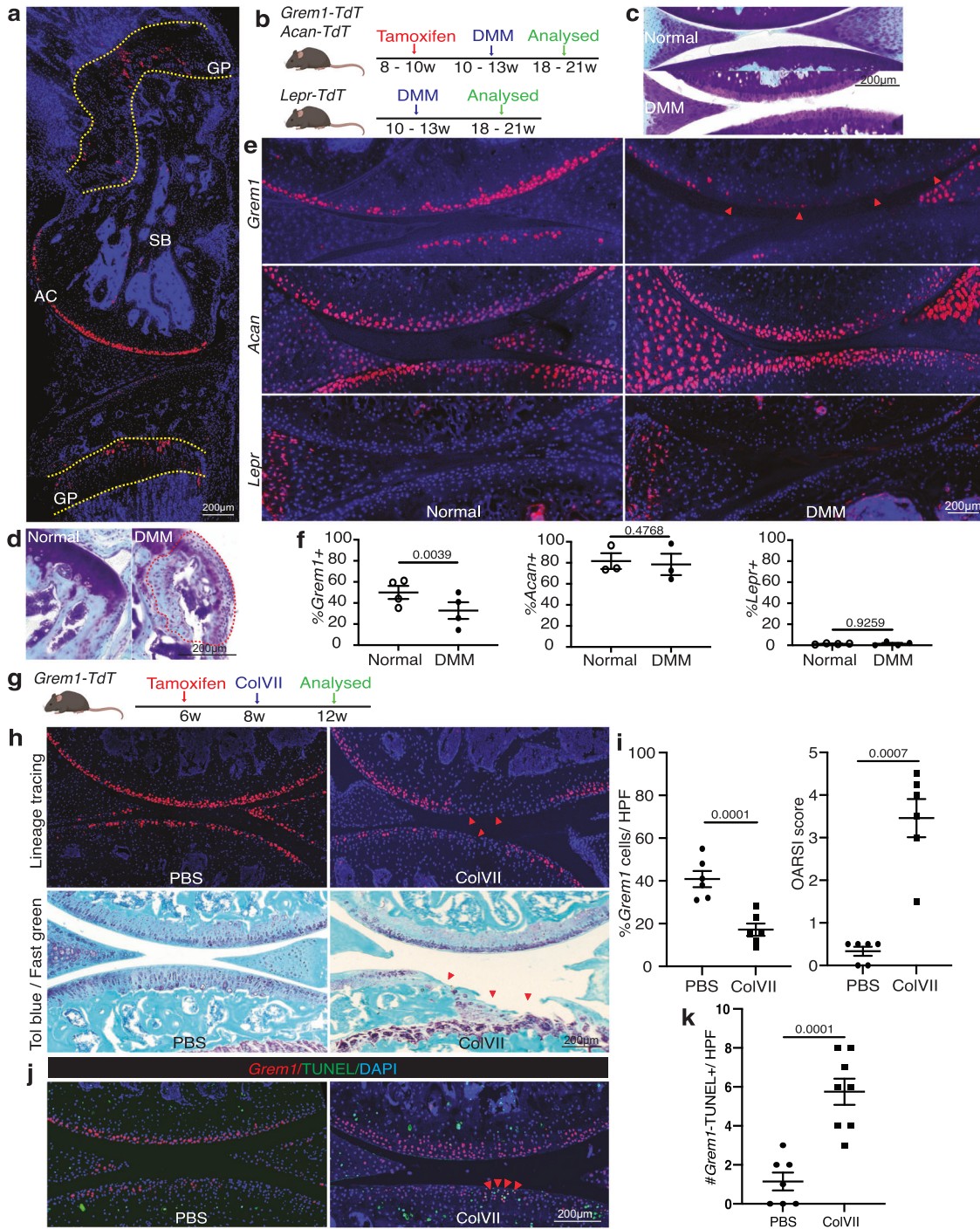

**Fig. 1 | The articular *Grem1*-lineage progenitor cells are significantly depleted in OA. a** Representative image of knee joint from 8-week-old *Grem1-TdT* mice administered tamoxifen at 6 weeks of age showing the location of *Grem1* cells in growth plate (GP), subchondral bone (SB) and articular cartilage (AC), *n* = 5 mice. **b** Experiment schema. DMM surgery was performed on adult *Grem1-TdT*, *Acan-TdT* and *Lepr-TdT* mice and tissue harvested after 8 weeks. **c** Representative image of proteoglycan loss, and **d** osteophyte-like formation (red dotted line) stained with Toluidine blue and Fast green in DMM with paired normal for comparison. *n* = 11 mice/group. **e** Representative images of paired distal femur joints of *Grem1-TdT* (top), *Acan-TdT* (middle) and *Lepr-TdT* (bottom) mice showing a decrease in *Grem1*-lineage cells within the AC DMM injury site (arrows) compared to normal. **f** Quantification of *Grem1*-, *Acan*- and *Lepr*-lineage cells as a percentage of total chondrocytes within the DMM injury site (filled circle) in comparison to no surgery (open circle) control. *Grem1* and *Lepr* *n* = 4 animals/group, *Acan* *n* = 3 animals/

group. Paired, two-tail *t*-test. **g** ColVII induced OA experiment schema. **h** Representative images of *Grem1-TdT* distal femur joints showing loss of *Grem1*-lineage AC cells as indicated by arrows within the injury site (top), and OA pathology induced by ColVII compared to PBS control. Sections stained with Toluidine blue and Fast green, arrows indicate superficial lesions. **i** Quantification of the percentage of *Grem1*-lineage AC cells per HPF (left) and, unblinded histopathological assessment using OARSI grading of OA pathology (right) in ColVII induced OA (square) compared to PBS controls (circle). *n* = 6 mice/group. Paired, two-tailed *t* test. **j** Representative images of TUNEL staining in articular *Grem1*-lineage cells quantified in **k**. **k** Quantification of the number of articular *Grem1*-lineage TUNEL positive cells in ColVII induced OA (square, *n* = 8 mice) compared to PBS control (circle, *n* = 7 mice). Unpaired, two-tailed *t* test. Bars denote s.e.m. Source data are provided as a Source Data file. Mouse image in figure schemas created with BioRender.com.

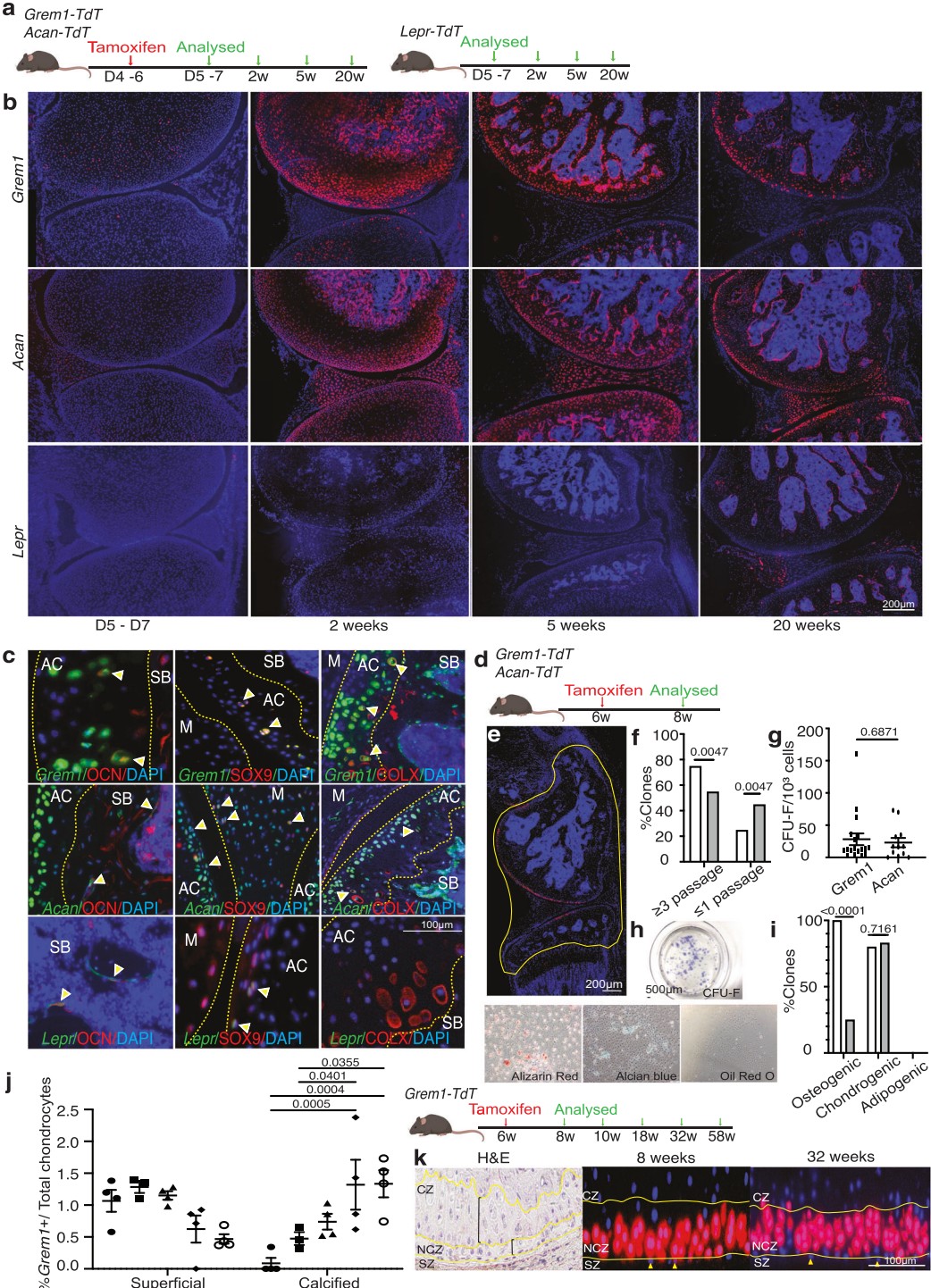

**Fig. 2 | *Grem1*-lineage marks a progenitor cell population in the AC.**
**a** Experimental schema. **b** Representative images of AC from *Grem1-TdT* (top row) and *Acan-TdT* (middle row) mice and age paired *Lepr-TdT* (bottom row) mice. *n* = 5 animals per group per time point. **c** Distal femur from neonatal tamoxifen *Grem1-TdT*, *Acan-TdT* and *Lepr-TdT* pulse-chased for 20 weeks. Representative IF staining of *Grem1* (top row), *Acan* (middle row) and *Lepr* (bottom row) cells expressing OCN, SOX9 and COLX (indicated by yellow arrows). Subchondral bone (SB), articular cartilage (AC) and meniscus (M). *n* = 3 mice per group. **d** Experimental schema. **e** Articular joints, outlined in yellow, were used to isolate red cells for in vitro assays. **f** Percentage of *Grem1*-lineage (white) clones able to undergo expansion compared to *Acan*-lineage (grey) clones. **g** Number of CFU-F formed per clone. **h**, Representative images of *Grem1*-lineage cells stained for CFU-F or differentiation markers Alizarin Red (osteo), Alcian blue (chondro) and Oil Red O (adipo). **i** Number of *Grem1*- (white) or *Acan*-lineage (grey) clones that had undergone

differentiation quantified as % of the total number of clones. **f**–**i** cells pooled from *n* = 3 animals, 22 clones per lineage. **j** Experimental Schema (right). Quantification of the total number of *Grem1*-lineage cells within the superficial and calcified zones as a percentage of the total number of chondrocytes within the AC at 8 weeks (filled circle, *n* = 4 mice), 10 weeks (square, *n* = 3 mice), 18 weeks (triangle, *n* = 4 mice), 32 weeks (diamond, *n* = 4 mice) and 58 weeks (open circle, *n* = 4 mice) of age (left). Bars show s.e.m. **k** Representative images of adult articular joint showing H&E of zonal organisation of chondrocytes in the superficial zone (SZ), non-calcified zone (NCZ) and calcified zone (CZ) and *Grem1*-lineage AC cells moving towards the CZ from 8 weeks (middle) to 32 weeks (right) of age. Superficial chondrocytes indicated with yellow arrows. **f**, **i** Two-sided Fisher's exact test. **g** Two-sided unpaired *t* test. **j** Two-way Anova Tukey's test. Bars denote s.e.m. Source data are provided as a Source Data file. Mouse image in figure schemas created with BioRender.com.

of osteogenic differentiation compared to *Acan*-lineage clones (100% vs 25%, <0.0001, Fig. 2i).

Normal AC development and maintenance requires both interstitial and appositional growth of the AC[7]. Using long-term cell fate tracing of adult *Grem1*-lineage cells in vivo, these cells contributed to progenitor populations in the superficial zone of the AC and with a significant increase in the percentage of *Grem1*-lineage cells within the calcified zone of the AC with age (Fig. 2j, k).

### *Grem1*-lineage articular CP cells are lost with age

To measure the longevity of *Grem1* lineage cells within the tibiofemoral joint, 6-week-old *Grem1-TdT* mice were administered tamoxifen in early adulthood and analysed during aging (Fig. 3a). There was a significant decrease in *Grem1*-lineage articular CP cells from young to aged adult mice (Fig. 3a, b). This is consistent with other studies looking at the presence and regenerative capacity of skeletal stem cells in aged animals[13,25]. The reduced regenerative capacity of the AC with age is due, at least in part, to reduced *Grem1*-lineage articular CP cells and chondrocyte proliferation (Supplementary Fig. 3a, b).

We next examined whether *Grem1*-lineage CP cells persist in the aging knee by administration of tamoxifen to *Grem1-TdT* mice at 29 weeks-of-age, in comparison to postnatal (week 1) or early adulthood (6 weeks), and visualisation of lineage traced cells 2 weeks later (Supplementary Fig. 3c). *Grem1*-lineage cell number decreased significantly with increasing adult age, with only <0.4% of total articular surface cells being *Grem1*-lineage cells in early middle-age (Supplementary Fig. 3d, e). At this stage, *Grem1*-lineage cells were mainly observed in the subchondral marrow space and occasionally in the retained, but no longer proliferating GP (Supplementary Fig. 3f).

### Targeted ablation of *Grem1* articular CP cells causes OA

The previous experiments confirmed that *Grem1*-lineage AC resident cells are bipotent chondro-osteogenic progenitor cells. To functionally test whether ablation interrupts the development of the AC we generated a new knock-in mouse line and employed other genetic ablation experiments. We had previously examined the role of *Grem1*-lineage cells in postnatal skeletogenesis using a diphtheria toxin ablation model (*Grem1-creERT;R26-LSL-ZsGreen;R26-LSL-DTA*), in which the *Grem1*-lineage was incompletely ablated[16]. Nevertheless, post-natal ablation of *Grem1*-lineage cells led to significantly reduced femoral bone volume by microCT after two weeks, with reduced trabecular bone as quantified by histology[16]. To achieve more complete ablation of *Grem1*-expressing cells, we generated a new *Grem1-DTR-TdTomato* knock in mouse model (*Grem1-DTR-Td*) in which *Grem1* actively expressing cells concomitantly express the TdTomato reporter and the Diphtheria toxin receptor (DTR), rendering these cells susceptible to diphtheria toxin (DT) ablation (Supplementary Fig. 3g).

*Grem1-DTR-Td* mice were administered 2 doses of DT intra-articularly, to target the articular population of *Grem1*-expressing cells, at 5 to 6 weeks of age and the mice were sacrificed 3 days later (Fig. 3c). This induced a significant reduction in *Grem1* CP cell number in the AC but not GP, increased COLX+ articular chondrocytes and induced OA pathological changes compared to age-matched controls (Fig. 3d–l, Supplementary Fig. 3h, i). The worsening in OA pathology scoring in the *Grem1*-cell ablation group was predominantly due to cartilage hypertrophy, proteoglycan loss and structural damage, with a lesser contribution from meniscus pathology and subchondral vascular invasion and zero scoring from subchondral bone sclerosis or osteophyte changes.

To confirm the role of adult *Grem1* CP cells in OA, we utilised a second transgenic mouse model of targeted cell ablation generated by mating *Grem1-TdT* mice to *Rosa-iDTR* mice to create *Grem1-TdTomato-iDTR* mice (*Grem1-creERT;DTR*). DTR expression on *Grem1*-lineage cells was induced by tamoxifen administration at 4 weeks of age, followed by local ablation of *Grem1*-lineage AC cells by intra-articular injection

of DT at 8-weeks-old for 2 weeks, before animals were sacrificed at 26 weeks (Supplementary Fig. 3j). Analysis of OA pathology and quantification of *Grem1*-lineage articular surface CP cells showed a significant loss in *Grem1*-lineage AC cells concomitant with worsened OA pathology, including a reduction in AC thickness (Supplementary Fig. 3k). Together our data confirmed that *Grem1*-expressing CP cells are normal progenitor cells that are lost in aging and in mechanical trauma-mediated OA and their depletion directly results in OA.

While we showed that loss of *Grem1*-expressing CP cells is an early event in OA (Fig. 1) and, in turn, also causes OA (Fig. 3 and Supplementary Fig. 3), we considered the possibility that OA may also be caused, in part, by the death or degeneration of the resultant *Grem1*-lineage, mature *Acan*-expressing chondrocytes. To first understand the degree by which *Grem1*-lineage CPs may be distinct from *Acan*-expressing articular chondrocytes, we undertook scRNAseq transcriptomic analysis of FACS-isolated single cells from early adult *Grem1-TdT* knee AC following tamoxifen administration (Fig. 3m). This analysis showed that while the majority (72%) of these *Grem1*-lineage traced cells expressed *Acan*, we could also identify *Grem1*-lineage cells that did not express *Acan* (Fig. 3m). This was consistent with co-immunofluoresence staining of knee sections in which double positive *Grem1*-lineage/ACAN+ cells were observed in the articular cartilage, but also discrete *Grem1*-lineage and ACAN expressing populations (Supplementary Fig. 3l).

To investigate potential differences between *Grem1* and *Acan* expressing AC cells, we identified significantly differentially regulated transcripts between *Grem1+Acan-* and *Grem1-Acan+* populations in our scRNAseq dataset (Supplementary Fig. 3m). The most highly upregulated transcript in *Grem1+Acan-* compared to *Grem1-Acan+* populations was *Islet1 (Isl1)*. *Isl1* encodes an essential LIM-homeodomain transcription factor important for progenitor populations and differentiation across multiple tissues including islet cells and pancreatic mesenchyme[26], motor neurons[27] and cardiovascular progenitor populations[28], but with an underexplored potential function in the AC. Together with differential expression of genes with roles in cartilage or arthritis, such as *platelet derived growth factor receptor alpha* (*Pdgfra*,[29]), *procollagen C-endopeptidase enhancer 2* (*Pcolce2*,[30]) and *hyaluronan and proteoglycan link protein 1* (*Hapln1*,[31]), we noted reduced expression of three collagen transcripts (*Col9a2, Col9a3, Col27a1*) in the *Grem1+Acan-* compared to *Grem1-Acan+* populations (Supplementary Fig. 3n). These transcripts encode key components of the type IX and XXVII extracellular matrix collagens, with vital roles in skeletal and cartilage development in mice and humans[32–34]. Altogether, this analysis suggested that *Grem1*-expressing articular CPs overlap with the broader *Acan*-expressing population, but are also distinct, and may have distinct functionality generated by differential expression of genes important for cartilage and progenitor populations.

Previous studies noted that ablation of chondrocytes marked by *Acan* or *Prg4* results in mild OA that resolves over time, or no OA[35,36]. To further discriminate the role of *Acan*+ chondrocytes in comparison to *Grem1*-lineage CPs in OA, we compared the OA phenotype caused by ablation of each cell population using similar genetic models and induction regimens. DT treatment of *Acan-creERT;DTR* mice caused a significant decrease in ACAN-expressing cells, and a mild but significant increase in OA pathology scoring at 26 weeks in comparison to PBS treated controls (Supplementary Fig. 3o, p). The overall average OA score in *Acan*-lineage ablated mice was 0.53 (+/− 0.27, st.dev.) in comparison to 1.75 (+/− 0.68) in the *Grem1-TdT-DTR* mice treated with DT (Supplementary Fig. 3k, o, p). These data are consistent with previous *Acan*- and *Prg4*- chondrocyte ablation experiments reported to generate mild or insignificant OA[35,36]. Given these differences, we note that OA is not just a disease of mature chondrocyte loss, but is likely also a failure of local progenitor regeneration.

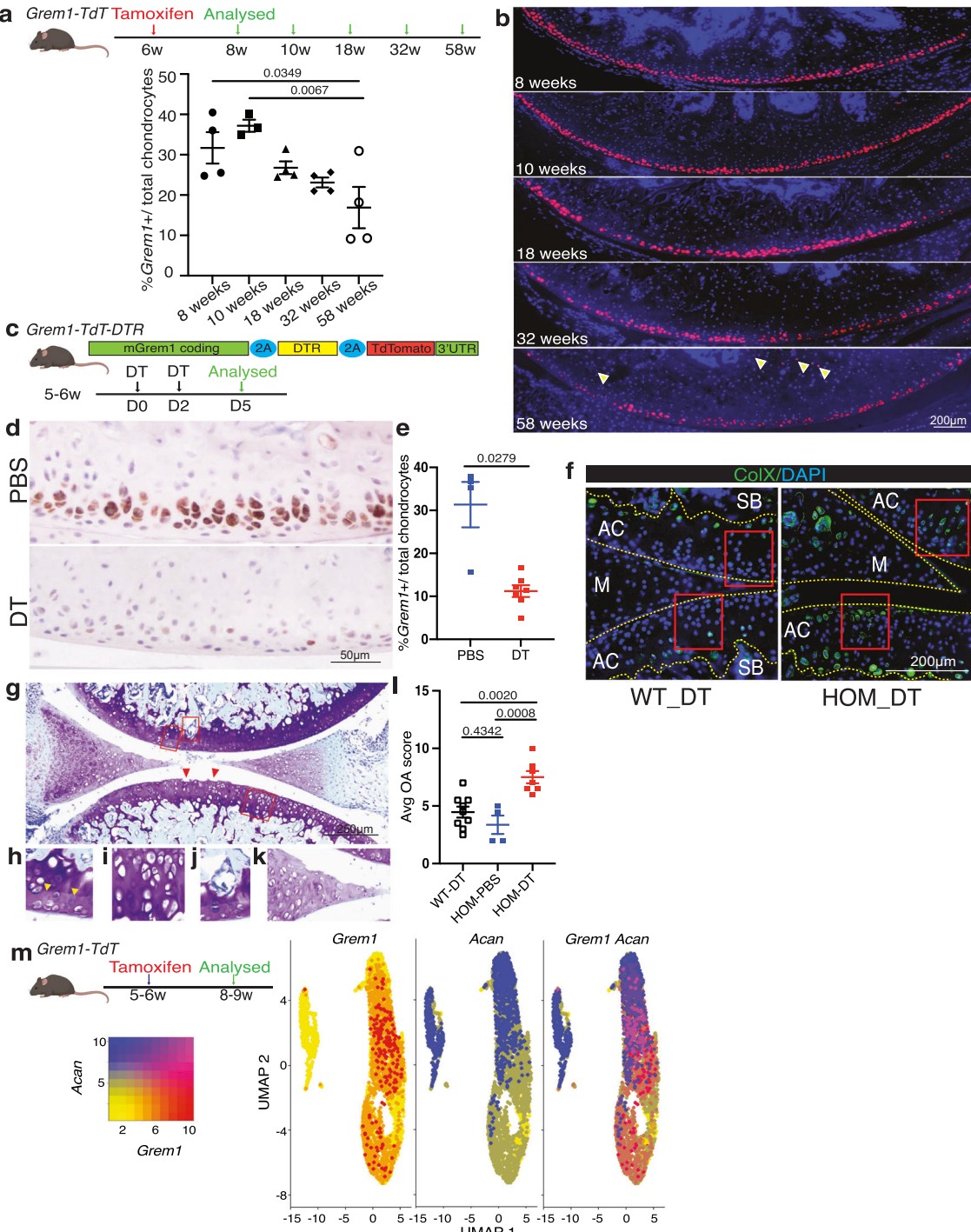

**Fig. 3 | _Grem1_-lineage articular progenitors cells are lost with age and targeted ablation causes OA. a** Experimental schema. Quantification of total _Grem1_-lineage articular chondrocytes (AC) as a % of the total ACs. $n = 4$ animals per time point, except 10wk $n = 3$. One-way Anova Tukey's test. **b** Representative images of the _Grem1-TdT_ articular knee joint from **a**. **c** Schematic of the _Grem1-Td-DTR_ knock-in construct (top) and experimental outline. **d** Representative images of early adult mice articular joints treated with PBS or DT stained with anti-RFP. **e** Quantification of _Grem1_ AC cells as a % of the total number of AC chondrocytes in PBS (blue, $n = 4$ mice) or DT treated (red, $n = 7$ mice) _Grem1-Td-DTR_ mice. Two-tailed Welch's _t_ test. **f** Representative images of ColX staining of _Grem1-Td-DTR_ joints from WT and HOM DT-treated animals, $n = 3$ mice/group. Boxed regions indicate loss of ColX chondrocytes following ablation of _Grem1_-expressing articular cells in HOM DT treated mice. **g** Representative image of DT treated articular joint of _Grem1-Td-DTR_ mice

($n = 7$ mice) stained with Tol blue and fast green showing pathological changes commonly associated with OA including AC damage indicated by red arrows, **h** loss of proteoglycan in the AC indicated by yellow arrows, **i** hypertrophic chondrocytes, **j** SB invasion, and **k** meniscus pathology. **l** Blinded scoring of average OA score following 3 days DT treatment in control wild-type mice (WT-DT, $n = 9$ mice), homozygous _Grem1-Td-DTR_ mice (HOM-DT, $n = 7$) or PBS _Grem1-Td-DTR_ mice (HOM-PBS, $n = 4$). One-way Anova Tukey's test. **m** Single cell RNA (scRNA) sequencing experiment schema for isolation of _Grem1_-lineage articular cells from mouse knee and analysis to show high _Grem1_ (red), high _Acan_ (blue) or high both _Grem1/Acan_ expressing (magenta) populations. Bars denote s.e.m. Source data are provided as a Source Data file. Mouse image in figure schemas created with BioRender.com.

We next used flow cytometry to isolate *Grem1*-lineage cells from early adult mice and implanted them into recipient mice that had undergone DMM surgery via intra-articular injection (Supplementary Fig. 3q). This initial effort to implant *Grem1*-lineage cells was hindered by inefficient homing to and/or survival of the cells in the AC (Supplementary Fig. 3r). Thus, we subsequently studied how to preserve this important population through normal aging (Fig. 3) or injury associated (Fig. 1) attrition that could be key for OA prevention.

### *Grem1*-lineage single cell transcriptomics revealed a distinct population of articular chondrocytes

We applied scRNAseq analysis to characterise FACS-isolated early adult *Grem1*-lineage CP cells from the epiphysis proximal to the GP and compared to *Grem1*-lineage cells within the GP, as well as age-matched skeletal cells defined by the *Lepr* lineage (Supplementary Fig. 4a, b). Unsupervised clustering of our pooled scRNAseq data revealed 6 distinct cell clusters: five *Grem1*-lineage (AC1-2, GP1-3) and one *Lepr* cluster (Fig. 4a, b, Supplementary Fig. 4c–e). Though there were substantial similarities in transcript expression in the articular surface and GP cells, clusters within the *Grem1*-lineage cells in the articular surface could also be separated by differential transcript expression from those in the GP (Fig. 4a, b, Supplementary Fig. 4f and Supplementary Table 1). Consistent with previous studies, this included enrichment of the AC marker, *Lubricin* (*Prg4*), in AC compared to GP *Grem1*-lineage cells[37] (Supplementary Fig. 4g).

Immunofluorescence staining of tissue samples from 2 week lineage traced *Grem1-TdT* mice showed overlap between articular *Greml*-lineage CP and PRG4 expressing cells (12.6% of the articular cartilage double positive), but the *Greml*-lineage cells were largely distinct from COL2-expressing chondrocytes in the calcified zone (<0.1% articular cartilage cells double positive for *Grem1*-lineage and COL2, Supplementary Fig. 4h). Most *Grem1*-lineage cells separately prepared from whole bone expressed the defining marker of mSSC, CD200[15] (Supplementary Fig. 4i), but only 19% of *Grem1*-expressing cells from the articular surface had a transcriptomic profile consistent with the mSSC immunophenotype (Supplementary Fig. 4j, k). This indicates that *Grem1*-lineage CP in the articular surface and mSSC in the GP partially overlap, but the articular surface *Grem1*-lineage population is distinct.

Next, we investigated gene expression associated with cartilage development and regeneration across the scRNAseq data set. *Forkhead box protein-o* (*Foxo*) genes have important roles in apoptosis, cell-cycle progression, resistance to oxidative-stress and maintenance of stem cell regenerative potential[38,39]. We observed that *Foxo1* expression was highly correlated with *Grem1* expression ($p < 0.0001$) in articular CP cells in our scRNAseq data[40] (Fig. 4c), with *Foxo1* expression also significantly increased in the AC compared to GP *Grem1*-lineage populations (Supplementary Fig. 5a). At the protein level, the majority of FOXO1 expressing cells in the adult AC were *Grem1*-lineage cells, while there were very few FOXO1 expressing *Grem1*-lineage cells observed in the GP (Supplementary Fig. 5b).

To determine whether *Foxo1* expression was essential for survival of *Grem1* articular CP cells and prevention of OA, *Grem1-TdT* mice were crossed to *Foxo1^{fl/fl}* mice, to produce *Grem1-TdT-Foxo1* conditional knock-out mice. *Grem1-TdT-Foxo1* mice were administered tamoxifen at 6 weeks of age to induce *Foxo1* deletion in *Grem1*-lineage cells and sacrificed at 26 weeks (Fig. 4d). Significantly fewer *Grem1*-lineage AC but not GP cells were observed in *Grem1-TdT-Foxo1* mice, in comparison to age-matched *Grem1-TdT* controls, and the resulting increased OA pathology resembled *Grem1-creERT;DTR* DT ablation (Fig. 4e–g, Supplementary Fig. 5c). This suggested that *Foxo1* was important to maintain adult *Grem1*-lineage articular CP cells and, by extension, AC integrity. Interestingly, *Grem1-TdT-Foxo1* mice administered tamoxifen in the early neonatal period (Day 4–6) developed drastic *Grem1*-lineage articular CP cell loss, more severe OA pathology and decreased AC

thickness, in comparison to age-matched *Grem1-TdT* controls and *Grem1-TdT-Foxo1* animals induced with tamoxifen in early adulthood (Supplementary Fig. 5d, e). This indicates the crucial role for *Foxo1* in AC development of the *Grem1*-lineage CP cell population, as well as in adult AC maintenance.

To discover molecular targets in *Grem1*-lineage articular CP cells for OA therapy, we focussed on fibroblast growth factor (FGF) signalling, given the role of this pathway in regulating chondrogenesis[41]. *Fibroblast growth factor receptor 3* (*Fgfr3*) was expressed both in *Grem1*-lineage GP and articular CP populations. Aware of the important role for fibroblast growth factor 18 (FGF18) as a key agonist of FGFR3 as an experimental treatment for OA[42], we investigated the impact of exogenous FGF18 on *Grem1*-lineage AC stem cells in vivo. Adult *Grem1-TdT* mice induced with tamoxifen were injected intra-articularly with 0.5 µg FGF18 or PBS twice weekly for 2 weeks and concurrently administered EdU to label newly proliferating cells (Supplementary Fig. 5f). FGF18 treatment resulted in a significant increase in both the number of EdU+ *Grem1*-lineage articular CP cells and AC thickness, suggesting FGF18 induces AC chondrogenesis in the adult knee via increased *Grem1*-lineage articular CP cell proliferation (Supplementary Fig. 5g, h). Notably, very few EdU+ cells were found in the hypertrophic chondrocyte zone, suggesting that the neo-cartilage arises from the articular CP population. To determine whether FGF18 can also rescue OA pathology, we used ColVII to induce OA and then intra-articularly injected *Grem1-TdT* mice with 0.5 µg FGF18 or PBS twice weekly for 2 weeks. FGF18 injection resulted in a significant increase in *Grem1*-lineage CP cells and AC thickness in treated joints compared to PBS controls and a significant reduction in OA pathology (Fig. 4h–j and Supplementary Fig. 5i). This suggested that exogenous FGF18 alleviates OA through increased *Grem1*-lineage CP cell proliferation.

## Discussion

OA, like other degenerative diseases, can be viewed as the dysfunction of mature cartilage cells or the result of an imbalance in stem-progenitor cell repair and renewal. Previous studies have shown functional cartilage cell dysfunction/death results in OA, here we show that ablation of a restricted progenitor population can cause OA without disturbing the broader population of differentiated chondrocytes.

While stem cell therapy for OA using traditional MSC populations has shown promise in reducing pain and stiffness, it does not effectively repair normal AC[43,44]. Several research teams, including our own, have recently described SSCs with lineage potential restricted to cartilage, bone and stroma that are promising candidates for joint disease treatment[13,15,16]. In this study, we show that *Grem1*-lineage CP cells contribute to neonatal generation of the AC and are pivotal to AC maintenance in adulthood. In vitro, the articular *Grem1*-lineage cells display clonogenicity and multi-potent differentiation properties associated with stemness, however, a complete assessment of their potential stem cell properties and lineage hierarchy using in vivo transplantation studies[15,45] remains to be undertaken.

The position of adult *Grem1*-lineage traced cells on the superficial surface of the AC, and subsequently in deeper layer chondrocytes and subchondral bone, is consistent with previous descriptions of chondrocytic label retaining progenitors and the *Prg4* (encoding lubricin) lineage where tracing began from birth or in juvenile mice, not adults as investigated here[7,37]. A key difference between the *Prg4*- and *Grem1*-lineages is that *Grem1*-expressing cells contribute to both articular and growth plate chondrocytes[37]. Although the AC and GP are both broadly constituted by chondrocytes, these structures form quite differently and have differing roles in skeletogenesis[46,47], with the AC being the primary site of OA pathology. Our focus has been on the articular *Grem1*-lineage population and superficial chondrocytes as this is the site of early OA-like changes. We cannot discount, however, that the GP *Grem1*-

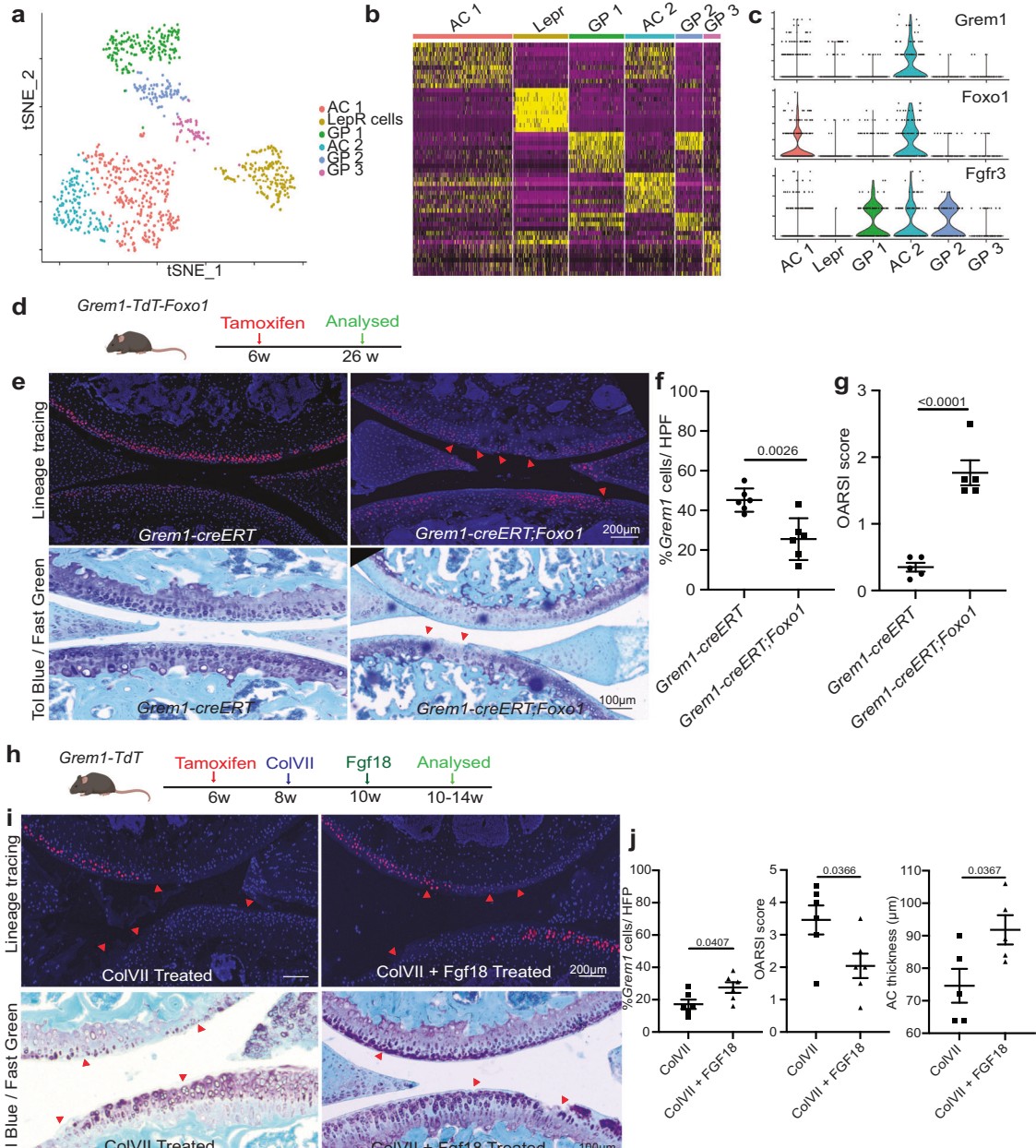

**Fig. 4 | *Grem1*-lineage single cell transcriptomics revealed a distinct population of articular chondrocytes that can be targeted for therapy. a** Single cell RNA (scRNA) sequencing data showed distinct clusters of cells isolated from the AC and GP of *Grem1-TdT* mice compared to *Lepr*-lineage cells isolated from the *Lepr-TdT* mice. **b** Heat map depicting unsupervised clustering of top 10 differentially expressed transcripts between the different clusters in **a**. **c** *Grem1*-lineage AC cells co-expressed genes important for AC function (*Foxo1*) and receptor (*Fgfr3*) for FGF18 treatment. One-sided chi-Square correlation analysis confirmed co-expression of *Foxo1* and *Fgfr3* in *Grem1* expressing cells (*p* = 0.00113). **d** Experiment schema. **e** Representative images of fluorescent *Grem1* lineage tracing in the articular joint with or without *Foxo1* deletion (top) with OA lesions highlighted using red arrows, and Toluidine blue and Fast green stain (bottom) with arrows indicating cartilage lesions and chondrocyte disorganisation. *n* = 5 animals per group. **f** Percentage of *Grem1*-lineage ACs per HFP in control *Grem1-TdT* mice (circle) compared to *Grem1-TdT-Foxo1* mice (square), *n* = 6 mice/group. **g** Unblinded histopathological OARSI scoring in *Grem1-TdT-Foxo1* mice (square) compared to *Grem1-TdT* mice (circle), *n* = 5 mice/group. **f, g** bars denote st.dev. **h** Experimental schema. **i** Representative images of joints from *Grem1*-lineage mice with ColVII induced OA with or without FGF18 treatment with arrows indicating injury site (fluorescence, top), toluidine blue and fast green stained showing proteoglycan loss and lesions indicated by arrows (bottom). **j** Quantification of the percentage of *Grem1*-lineage cells with FGF18 treatment (left, *n* = 6 mice/group), OA score (middle, *n* = 6 mice/group) and AC thickness (right, *n* = 5 mice/group). Bars denote s.e.m. **f–j** unpaired, two-tailed *t* test. Source data are provided as a Source Data file. Mouse image in figure schemas created with BioRender.com.

lineage CP cells may also contribute to the articular phenotypes observed and have not categorically excluded that *Grem1*-lineage subchondral bone populations may arise in part from GP resident progenitors. Equally, we have not investigated which *Grem1*-lineage population gives rise to the meniscus or if the *Grem1* promoter is independently active at sites throughout the knee joint.

We show that some, but not all, acutely labelled *Grem1*-lineage articular cells express *Prg4*/PRG4 and/or Acan, and are distinct from the COL2-expressing chondrocytes of the calcified zone (Supplementary Fig. 3l, 4h). Other key differences between the *Grem1*-lineage and that of the broader chondrocytic population marked by *Acan* include the extent and location of labelled lineage cells (Fig. 2e, Supplementary

Fig. 2b), differential expression of cartilage-related transcripts (Supplementary Fig. 3n), clonogenicity and differential potential ex vivo (Fig. 2d–i) and response to injury (Fig. 1b–f), such that *Grem1-* but not *Acan-*lineage cells were lost in the early OA-like phenotype induced by DMM surgery. Ablation of *Grem1-*lineage CP cells in the knee joint causes OA, with a milder phenotype observed using similar induction protocols and transgenic mouse models to ablate *Acan-*lineage cells (Supplementary Fig. 3k, o). Likewise, as FOXO1 expression is restricted to the superficial chondrocytes of the AC (Supplementary Fig. 5b), genetic deletion of *Foxo1* in *Grem1-*lineage cells effectively depletes FOXO1 expression in the articular surface, rather than deeper chondrocytic layers and results in OA. This suggests the *Grem1-*expressing cells represent a specific subset of the total chondrocyte population that may be lost early in the disease process.

We acknowledge that OA can be caused by dysfunction of mature cartilage cells, but here also highlight the importance of dysfunction and loss of a specific chondrocyte progenitor population marked by *Grem1*. This is supported by phenotypic differences resulting from adult *Prg4* and *Grem1-*lineage ablation. Loss of both *Prg4-* and *Grem1-* lineage populations resulted in decreased superficial chondrocytes, however cartilage deterioration and OA histological features only occurred following loss of *Grem1-*, and not *Prg4-*, lineages[36] (Fig. 3).

Similar to mSSC cells[13], *Grem1-*lineage articular CP cells are lost with age. Using two mouse models of OA, we found that *Grem1-*lineage CP cells in the knee joint were lost in early injury. We also report that functional ablation of *Grem1-*lineage articular CP cells, or genetic deletion of *Foxo1* in *Grem1-*lineage cells, in early adult mice led to significant and rapid OA (Figs. 3, 4). As efforts to reintroduce *Grem1* articular CP cells would be a natural first step to therapeutics, our initial efforts were not successful (Supplementary Fig. 3q) and future efforts using a bio- scaffold with factors to stimulate enhanced chondrogenesis are warranted. scRNAseq analysis identified separate populations of *Grem1-*lineage cells within the AC and GP structures, that provided an insight to the restricted commitment of each population of cells. FGF18 treatment induced proliferation of *Grem1-*lineage articular CP cells and reduced OA pathology, presenting a strong candidate for further clinical trials of OA prevention and therapy for early disease.

We propose that OA can be predisposed by inadequate reserves of, or injury to, articular CP cells (accompanied by the loss of surface proteoglycan and cartilage fibrillation), followed by the apoptotic death of further CP cells, and the resulting failure to regenerate articular cartilage. This study reframes OA as a cellular disease that can result from CP cell loss and provides a biological focus for OA therapy.

## Methods

### Mice
*LepR-cre*[48], *Acan-creERT*[49], *Grem1-creERT*[16], *R26-LSL-TdTomato*[50], *R26-LSL-ZsGreen*[50] and *R26-LSL-iDTR* were imported from Jackson Laboratory and bred within the South Australia Health and Medical Research Institute (SAHMRI) Bioresources facility and Institute of Comparative Medicine at Columbia University. *Grem1-TdTomato-iDTR* mice (*Grem1-creERT;DTR)* were generated by mating homozygous *Grem1-TdT* mice to homozygous *Rosa-iDTR*. *Acan-iDTR* mice (*Acan-creERT;DTR)* were generated by mating heterozygous *Acan-creERT* mice to homozygous or heterozygous *Rosa-iDTR*. All animal experiments complied with all relevant ethical regulations and were approved by the Animal Ethics Committee at the SAHMRI under ethics number SAM189 and SAM385.19 and at Columbia University.

Mice were maintained in the South Australia Health and Medical Research Institute (SAHMRI) Bioresources facility and Institute of Comparative Medicine at Columbia University, in accordance with JAX USA animal husbandry protocols. Animals were given food and water ad libitum and housed in temperature- (average 21–22 °C), moisture- (45–55% humidity), and light-controlled (12 h light/dark cycle)

individually ventilated cage systems. 10–13 weeks old *Grem1-TdT, Acan-TdT, LepR-TdT* and *C57BL/6* male mice were used to examine lineage tracing in health and DMM surgery model. Postnatal Day 4–6 (neonatal) and 6 weeks old (adult) *Grem1-TdT, Acan-TdT* and *LepR-TdT* mice were used to examine lineage tracing in early development and adulthood. 8 weeks old *Grem1-TdT* mice were given tamoxifen at 6 weeks of age to examine in vitro stem cell properties. 26-week-old *Grem1-creERT;FOXO1* mice were given tamoxifen at postnatal Day 4–6 (neonatal) and 6 weeks of age to examine the AC impact of *FOXO1* deletion in *Grem1* cells. 5–6 week old *Grem1-TdT* mice were given tamoxifen for ten days, then harvested at 7–9 weeks of age for scRNAseq analyses. 4–6 week old *Grem1-TdT* mice were used to examine the treatment of Fgf18 in AC with or without CiOA 4–6 week old *Grem1-TdT* mice were used to examine the treatment of Fgf18 in AC with or without CiOA. 4–7 week old *Grem1-Td-DTR, Grem1-creERT;DTR* and *Acan-creERT;DTR* mice were used to examine the impact of the ablation of *Grem1-*lineage and *Acan-*lineage cells on AC.

### Generation of knock-in *Grem1-DTR-TdT* mice by CRISPR/Cas9 mediated genome engineering
To generate a DNA construct for Diphtheria Toxin Receptor (DTR) / tdTomato (tdTom) expression under the control of the endogenous *Grem1* promoter in mice, we first used gene synthesis to insert the coding sequence for DTR immediately downstream of a 2 A peptide from porcine teschovirus-1 polyprotein (P2A), followed by a second P2A sequence and the coding sequence for tdTom, all within two exon2 2 kb *Grem1* homology arms (Supplementary Fig. 6). The construct was Sanger sequence verified (Geneuniversal) and used for subsequent gene targeting in mice. C57BL6/J female mice were superovulated with 5 IU Pregnant Mare Serum Gonadotrophin (PMSG; Folligon; Intervet) and 47.5 h later with 5 IU human chorionic gonadotrophin (hCG; Chorulon; Intervet) before being mated to C57BL6/J males. Presumptive zygotes were collected from oviducts 19 h post-hCG injection in FHM media (Merk Millipore) and maintained in KSOMAA media (Merk Millipore) under oil at 37 °C in 5% $CO_2$ 5% $O_2$ with a Nitrogen balance. Fertilized zygotes were identified in FHM media under oil by the presence of two pronuclei before microinjection of 25 ng/ul SpCas9 protein (PNA Bio), and 10 ng/ul SpCas9 single guide RNA (sgRNA) (CGACTTGGATTAAGTCAAAG) with the donor plasmid at 10 ng/ul, in a buffered solution. Genomic DNA was isolated from ear notches using a High Pure PCR Template Preparation Kit (Roche) and Knock-in (KI) founder screening was performed using a forward primer (GCCTGGGATATTCTGAGCCC) and a reverse primer (GTCCCGCGCATTTTGACTTT). KI allele in the founder mouse was confirmed by PCR and Sanger sequencing across the entire locus from outside of the donor homology arms before mating with C57BL6/J to create second generation line.

### Tamoxifen administration
Neonatal *Grem1-TdT* and *Acan-TdT* mice were subcutaneously administered 2 mg tamoxifen (#T5648, Sigma) dissolved in peanut oil at Day 4–6 of age for lineage tracing studies. *Grem1-TdT* and *Acan-TdT* adult lineage tracing mice were administered 4 doses of 6 mg tamoxifen dissolved in peanut oil via oral gavage at ages specified in figures. *Grem1-TdT* and *Acan-TdT* DMM OA mice were administered 4 doses of 6 mg tamoxifen dissolved in peanut oil via oral gavage at 8–10 weeks of age. *Grem1-TdT* ColVII OA and *Grem1-creERT;DTR* mice were administered 5 separate doses of 4 mg tamoxifen via daily oral gavage over 9 days (1 day off dosing in between) from 6 or 4 weeks of age, respectively. EdU labelling of *Grem1-TdT* mice was performed by administering 5 separate doses of 4 mg tamoxifen via daily oral gavage over 9 days and commenced on EdU in drinking water 2 weeks prior to collection. *Grem1-TdT* mice for scRNA experiments and ACAN co-immunofluorescence staining were administered 500 mg/kg tamoxifen chow for 10–14 days from 5–6 weeks of age.

## Diphtheria toxin (DT) administration

*Grem1* cells within the AC are abundant at 2 weeks of age, therefore, we selected mice at this age for initial validation of *Grem1* cell ablation within the AC. Diphtheria toxin (#BML-G135, Enzo) at a concentration of 250 ng in 10 μL of PBS was injected intraperitoneally into 2-week-old *Grem1-Td-DTR* and mice sacrificed 2 days later. Anti-RFP immunohistological staining was performed to validate the distribution and ablation of *Grem1* cells in the *Grem1-Td-DTR* AC compared to aged matched *Grem1-TdT* mice (Supplementary Fig. 3g). For OA pathology, mice 5–7 weeks of age were administered two doses of 250 ng DT intra-articularly in 10 μL of PBS and limbs collected 3 days later for analysis and blinded scoring of OA pathology. *Grem1-Td-DTR* mice administered with DT developed a fatal small intestinal phenotype at day 4 for homozygous mice and at day 7 for heterozygous mice after DT injection. A similar phenotype has been previously reported[51]. Ablation of *Grem1*-lineage cells in *Grem1-creERT;DTR* mice and *Acan*-lineage cells in *Acan-creERT;DTR* mice was performed by administering 100–125 ng of DT (#D0564, Sigma) intra-articularly 3 times weekly for 2 weeks into 8 week old mice that were given tamoxifen prior at 4–6 weeks of age. Mice were then sacrificed at 26 weeks of age for analysis and unblinded OARSI scoring.

## Destabilisation of the medial meniscus surgery (DMM)

DMM surgery was conducted using an established protocol[52] approved by the SAHMRI Bioresources Animal Ethics Committee. Briefly, surgery was performed on male mice between the ages of 10–13 weeks that had previously been administered tamoxifen. Animals underwent general anaesthesia via isoflurane inhalation. The right hind limb of the animal was shaved with a razor blade to provide a clean surface and left hind limb was used as a paired normal control. Skin incision was made on the medial side of the patella, using sterile scissors, to expose the patella. Once the patella was exposed, a second incision was made on the medial side and the incision site was extended so that the patella could be luxated. Using sterile gauze, the fat pad located on the knee was pushed aside to expose the medial meniscotibial ligament, the ligament was then transacted using a twisting motion. Bleeding was controlled throughout surgery, and the cartilage was kept moist with sterile saline. The patella was then repositioned back, and the incision sites and patella were sutured.

## Collagenase VII induction of OA (CiOA)

The collagenase-induced OA model involves intra-articular injection of Type VII Collagenase as described[22]. Briefly, 8-week-old mice previously administered with tamoxifen were injected intra-articularly with 10 μL of 2U ColVII (#C0773, Sigma-Aldrich) dissolved in PBS under a dissection microscope.

## Histology

Mice were humanely sacrificed, bones from both hind limbs were collected and all muscles removed before fixing in 4% paraformaldehyde overnight, decalcified in Osteosoft® (#101728, Millipore) for 3–10 days and dehydrated in 30% sucrose at 4 °C before embedding in OCT compound (Sakura Tissue-Tek). Embedded tissues were stored at −80 °C. 10 μm frozen sections were collected on cryofilm (type IIC(10), Section-Lab) for staining. 0.04% toluidine blue (#198161, Sigma) in 0.1 M sodium acetate pH4.0 or 0.1% Safranin O (#8884, Sigma) in water, and 0.1% fast green (#F7252, Sigma) stain in MilliQ water was used to demonstrate histological features of cartilage and bone. Haematoxylin and eosin (H&E) staining of the AC region was used to identify different organisational zones within the adult AC.

## Quantification of cells and AC thickness

Lineage-traced (*TdTomato* positive) cells within the AC were counted throughout the femur and represented as a percentage of the total number of chondrocytes (DAPI positive). Only *TdTomato* cells colocalised with DAPI were included to avoid the inclusion of dead cells. AC cells in neonatal tracing were defined by the thin and highly cellular structures consisting of small and closely bound cells. Serial sections stained with H&E were used to guide the zonal separation of the AC. Cells in both the femur and tibia AC in 3 randomly selected images at 40x magnification per sample were used to quantify the number of lineage-traced cells per high power field (HPF) and represented as a percentage of the total number of chondrocytes. AC thickness was measured by averaging 3 randomly selected areas of the AC in the femur and tibia. Final AC thickness was represented as an average of 3 slides per sample. EdU and TUNEL positive *Grem1*-lineage cells were quantified in 3 randomly selected images at 40x magnification per sample and represented as total number of positive cells per HPF. Only EdU positive cells within 1-4 cells from the superficial zone of the articular layer were counted.

## OA pathology scoring

OA pathology (both femoral and tibial) in *Grem1-creERT;DTR* mice were scored using histological sections in an unblinded fashion using the 0–6 OARSI scoring system[23]. Final OARSI scores represent the average of 3 slides per sample. OA pathology in the *Grem1-TdT-DTR* experiments were scored blinded using scoring parameters of AC damage (0–11), proteoglycan loss (0–5), chondrocyte hypertrophy (0–3), subchondral bone sclerosis and vascular invasion (both 0–3), osteocyte size and maturity (both 0–3) and meniscus pathology (0–20) as described[53]. The average scores of 2–3 slides per sample, total of 8 parameters, were summed to produce the final score. The average OA scores of both tibia and femora were included in this study.

## Immunohistological and fluorescent staining

Immunohistochemistry and immunofluorescent staining were completed on 10μm frozen sections. Antigen retrieval was performed by placing slides in a steamer submerged in antigen unmasking solution (#H-3300, VectorLab) for 6 min. 0.025% v/v triton X-100 in PBS was used for anti-RFP staining. Immunohistochemistry slides were treated for endogenous peroxidase activity by incubating in 3% $H_2O_2$ for 30 min. Blocking was performed in 2% BSA, 5% normal goat and 5% normal donkey serum. The following antibodies were used: anti-PCNA (#ab18197 Abcam, 1:200), ColX (#ab58632 Abcam 1:200), OCN (#ab93876 Abcam, 1:200), Lubricin/PRG4 (#ab28484 Abcam, 1:250), COL2 (#ab34712 Abcam, 1:250), FOXO1 (#MA5-32114 ThermoFisher Invitrogen, 1:150), Sox9 (#AB5535 Millipore, 1:400), ACAN (#AB1031 Merck Millipore, 1:100) and anti-RFP (#600-401-379, Rockland 1:250). After overnight incubation at 4 °C, slides were washed with PBST and incubated with species-appropriate secondary antibody (1:200–1:300) at room temperature for 1 h. Finally, slides were counter stained with DAPI before mounted with a cover slip. For immunohistochemistry, after overnight incubation at 4 °C, slides were washed with PBST and incubated with anti-rabbit biotin (#BA-1000, VectorLab 1:250) at room temperature for 1 h and then streptavidin-HRP (#SA-5004, VectorLab 1:100) at room temperature for 30 min and developed with DAB chromogen (#K3468, Dako). EdU staining was done using the Click-iT EdU proliferation kit (#C10086, ThermoFisher) as per manufacturer's instructions.

## Imaging

Stained sections were scanned using the 3DHistech Panoramic 250 Flash II to generate brightfield images. Fluorescent images were captured either on a Olympus IX53 and Nikon Eclipse Ti inverted microscope or the Leica TCS SP8X/MP confocal microscope. Images of whole bone were achieved by stitching together 20–24 images at 10x magnification using ImageJ software.

### *Grem1* Chondroprogenitor Cell isolation

Both hind limb bones (femur and tibia) were collected from 8-week-old mice that had been subjected to tamoxifen and all muscles and ligaments removed. To limit contamination from non-articular cell populations present in whole bone preparations, the knee joints were dislocated at the epiphyses from both femur and tibia to exclude the GP-cartilage, gently disrupted using a mortar and pestle, and digested in 4 mg/ml collagenase IV (#17104019, Gibco) and 3 mg/ml Dispase (#17105041, Gibco) in α-MEM (#M5650, Sigma). Collected cells were then sorted for lineage-traced articular *TdTomato* positive cells. Cells were sorted based on forward and side scatter, single cells as well as positive for *TdTomato* fluorescence. Live and dead (DAPI) staining was omitted to allow for later expansion of clones in DAPI free conditions. Sorted cells were plated in complete media (α-MEM supplemented with 20% defined bovine serum, 100 mM L-ascorbate-2-phosphate, 1 mM sodium pyruvate, 50 μg/ml streptomycin, 50 U/ml penicillin, 2 mM L-glutamine and 10 μM Y-27632) at 1000 cells per 10 cm dish to allow for isolation of individual clones using clonal cylinders (#Z370789, Sigma). A total of 22–24 clones (i.e., isolated adherent clusters of >50 cells at 10–14 days) were harvested and allowed to expand before performing clonogenicity and multilineage differentiation assays. Both clones that expanded and did not expand were collected for qPCR analysis.

### Clonogenicity and multilineage differentiation

Colony-forming unit−fibroblasts (CFU-F) assay was performed by seeding cells at clonal density (1000 cells per 10 cm dish) and subsequently cultured in complete media for 14 days. The cultures were stained with 0.1%w/v toluidine blue in 2.4% formalin solution and the total number of colonies counted, where an individual colony was defined as ≥50 cells. The number of clones was reported as (CFU-F)/1000 cells plated. Induction of osteogenic, chondrogenic and adipogenic differentiation was conducted by maintenance of cell cultures in differentiation media (StemPro) as per manufacturer's instructions. Positive and negative assessment of differentiation potential was performed by staining cells with 2% w/v Alizarin red in water pH 4 (osteogenesis), 0.1% w/v Alcian blue in 0.1 N HCl (chondrogenesis) and 0.5% w/v Oil Red O in isopropanol diluted further in a 6:4 ratio in water (adipogenesis). All in-vitro assays reported in this study were conducted on clonal cell populations of passage 5 or lower.

### RNA isolation and RT-PCR

Total RNA was isolated from cells using TRIzol (Thermofisher) as per manufacturer's instructions. Complementary DNA (cDNA) was generated using SuperScript IV reverse transcriptase kit (Invitrogen) according to manufacturer's protocol. Transcript levels were assessed by QuantStudio 7 (Thermofisher) using IDT probes GAPDH (Mm.PT.39a.1) and Grem1 (Mm.PT.53a.31803129) with the FastStart TaqMan Probe Master (#04673409001, Roche).

### Flow Cytometry for assessment of mSSC markers in *Grem1*-lineage cells

All limbs were collected from 8-week-old mice that had been subjected to tamoxifen and all muscles and ligaments removed. Cleaned bones (i.e., whole bone preparation) were gently disrupted using a mortar and pestle, minced, and digested in 2.5 mg/mL collagenase I (#CLS-1, Worthington). Collected cells were subjected to RBC lysis and blocked with 3% BSA, 2% FCS in PBS. The following antibodies were used: anti-CD45 (#103131, BioLegend, 1:80), anti-Ter119 (#116227, BioLegend, 1:80), anti-CD31 (#562861, BD Pharmingen, 1:80), anti-CD200 (#ab33735, Abcam, 1:10), APC/Cy7 Streptavidin (#405208, BioLegend, 1:300) and DAPI (#D9542, Sigma-Aldrich). Cells were sorted on FACS-Fusion or LSR Fortessa (BD Biosciences) based on forward and side scatter plots, single cells, live cells (DAPI negative), trilineage expression (CD45⁻Ter119⁻CD31⁻) and lineage reporter positive fluorescence.

### Single cell sorting for scRNAseq analysis

*Grem1-TdT* mice at 5–6 weeks of age were subjected to tamoxifen chow for 10–14 days before they were sacrificed. Age paired *LepR-TdT* mice were used for the analysis presented in Fig. 4. Hind limbs were collected, and all muscles and ligaments removed. AC alone (Fig. 3), or AC and GP (Fig. 4) from *Grem1-TdT* mice were separately excised under a dissection microscope and kept as separate tissue samples (AC and GP) through subsequent mincing using a scalpel before being digested in 2.5 mg/mL collagenase type II (#CLS-2, Worthington). Cleaned *LepR-TdT* whole bones were gently disrupted using a mortar and pestle, minced, and digested in 2.5 mg/mL collagenase type I (#CLS-1, Worthington). The following antibodies were used: anti-CD45 (#103111, BioLegend), anti-Ter119 (#116211, BioLegend), anti-CD31 (#102509, BioLegend), and DAPI (#D9542, Sigma-Aldrich). *LepR* cells were sorted based on forward and side scatter, single cells, live cells (DAPI negative), trilineage negative expression (CD45⁻Ter119⁻CD31⁻) and TdTomato positive fluorescence. *Grem1* cells from the AC or GP (GP Fig. 4 only) were sorted separately based on forward and side scatter, single cells, live cells (DAPI negative), and TdTomato positive fluorescence, to generate separate *Grem1*-lineage populations derived from the AC (or GP). Individual cells were randomly allocated into 96-well plates to avoid batch effects, in lysis buffer previously prepared by the Sulzberger Genome Centre (Columbia University, Fig. 4) or encapsulated in Gel Beads-in-emulsion (GEMs) using a Chromium Controller (10× Genomics, Pleasanton, CA, USA, Fig. 3). Library preparation was performed using Chromium Single Cell 3′ Library & Gel Bead Kit v3 (10× Genomics, Pleasanton, CA, USA) following the manufacturer's protocol and sequenced using Illumina NextSeq 500 at Columbia University Genome Centre's Single Cell Analysis Core.

### Single cell sequencing analysis

Raw Illumina data were processed to convert Illumina basecall files to FASTQ format. Alignment of reads to the mouse genome (mm10) version 2020-A and gene counting was performed using the 10× Genomics Cell Ranger pipeline (10× Genomics, Pleasanton, CA, USA). Cells from 2–3 individual samples were grouped together. Count tables were analysed in the R environment (version 4.0.2 or 4.3.1) with the Seurat package (version 3.2.2 or v5). Figures 3 and 4 experiments and analyses were undertaken separately with analyses as follows.

Figure 3: Cells with fewer than 350 detected genes or genes that were expressed by fewer than 5 cells were excluded from the analysis. Dead cells and doublets were also removed by excluding cells that had more than 5% reads derived from mitochondria and cells with >5000 features. After removing all the unwanted cells from the dataset, the data was normalised by employing a global-scaling normalisation method 'LogNormalize'. Subsequently, the 2000 most variable genes were identified, the data were scaled, and the dimensionality of the data was reduced by principal component analysis (PCA). A K-nearest-neighbour (KNN) graph based on the Euclidean distance in PCA space was constructed using the 'FindNeighbors' function and applied Louvain algorithm to iteratively group cells together by the 'FindClusters' function. A non-linear dimensional reduction was then performed via uniform manifold approximation and projection (UMAP) and various cell clusters were identified using a resolution of 0.6. Marker genes per cluster were calculated using Seurat's 'FindAllMarkers' function and the 'wilcox' test option. To identify cells that express *Acan* and *Grem1*, a subset of cells were extracted based on normalised and transformed expression of *Grem1* and *Acan* with positive cells showing expression >0.25 and negative cells showing <0.25. Significantly differentially expressed genes between *Grem1-/Acan+* and *Grem1 + /Acan-* cells were identified using Seurat's 'FindMarkers' function and the 'wilcox' test option.

Figure 4: all lineage traced cells were combined into one single cell object. When using FindClusters() function, dimensions of reduction were set from 1 to 9 and resolution was set at 0.6 to capture

the majority of variation in the data (Supplementary Fig. 4c). This unsupervised clustering identified 6 distinct cell clusters. We undertook quality control assessments to check that sequencing depth (indicated by unique molecular identifiers per cell), sample quality (mitochondrial and ribosomal content per cell) and pro-liferation markers were not drivers of cluster identity (Supplementary Fig. 4d). Of the *Grem1*-lineage cells, AC cluster 1(AC1) showed differential expression of a cytoprotective marker, *Clusterin*[54], and early chondrogenic differentiation marker, *Integral membrane protein 2A*[55]; AC cluster 2 (AC2) the hypertrophic chondrocyte marker *Col10a1*[56] and early chondrocyte marker *Sox9*[57]; GP cluster 1 (GP1) and 2 (GP2) the cartilage angiostatic factor *Chrondromodulin-1 (Cnmd)*, known for its role in endochondral ossification and bone formation[58,59], melanoma inhibitory factor (*Mia*)[60] and *Matn3* which encodes an ECM protein that regulates cartilage development and homeostasis[61] while GP1 also expressed cartilage differentiation marker *Serpina1*[62]; GP cluster 3 (GP3) expressed cartilage degeneration marker *Mmp13*[63] (Fig. 4a, b).

To explore single cell clusters that co-express *Grem1*, *Foxo1* and *Fgfr3 a g*ene count greater than 0 was used to define positive expression, and Chi-Square statistics were employed. For cluster analysis, FindAllMarkers() was used to generate marker lists, with parameters of min.pct = 0.25, logfc.threshold = 0.25, and tes.use = ' roc'. To investigate the relationship between *Grem1* AC cells with previously published mouse skeletal stem cells[15](mSSCs; AlphaV+CD200+CD45−6C3−CD105−Ter-119−Tie2−Thy−), expression of the following combination of transcripts was investigated in the *Grem1* AC population. Expression of *Itga5* (encoding AlphaV) and *Cd200* (encoding CD200) and lack of expression of *Ptprc* (encoding CD45), *Enpep* (encoding 6C3), and *Eng* (encoding CD105). Transcripts for *Ly76* (encoding Ter119), *Tek* (encoding Tie2) and *Thy1* (encoding Thy1.1/Thy1.2), were not detected in the AC populations and so were not included in the mSSC transcriptomic selection criteria. All violin plots were generated with VlnPlot() function in Seurat package and Chi-Square tests were performed with chisq.test() function in the base package.

## Statistical analysis

All analyses were performed using Prism 8/9 (GraphPad Software Inc.), individual test details are provided in Figure legends. Line in dot plots indicate mean.

## Reporting summary

Further information on research design is available in the Nature Portfolio Reporting Summary linked to this article.

## Data availability

The scRNAseq data generated in this study have been deposited in the GEO database under accession codes GSE193175 and GSE242732. Grem1-DTR-TdT mice are available from S.L.W. under a material transfer agreement with SAHMRI. Source data are provided with this paper.

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

## Acknowledgements

We acknowledge the facilities and the scientific and technical assistance of the South Australian Genome Editing (SAGE) Facility, the University of Adelaide, and the South Australian Health and Medical Research Insti-tute. SAGE is supported by Phenomics Australia. Phenomics Australia is supported by the Australian Government through the National Colla-borative Research Infrastructure Strategy (NCRIS) programme. Flow cytometry analysis and/or cell sorting was performed at the Adelaide Health and BioMedical Precinct Cytometry Facility. The Facility is gen-erously supported by the Detmold Hoopman Group, Australian Cancer Research Foundation and Australian Government through the Zero Childhood Cancer Programme. We also acknowledge the facilities and scientific and technical assistance of the JP Sulzberger Columbia Gen-ome Centre and the Confocal and Specialised Microscopy Shared Resource of the Herbert Irving Comprehensive Cancer Centre at Columbia University, USA. This research was funded in part through the

NIH/NCI Cancer Centre Support Grant P30CA013696 and used the Genomics and High Throughput Screening Shared Resource. This publication was supported by the National Centre for Advancing Translational Sciences, National Institutes of Health, through Grant Number UL1TR001873. The content is solely the responsibility of the authors and does not necessarily represent the official views of the NIH. Mouse image in figure schemas created with BioRender.com. This study was supported by grants from the National Health and Medical Research Council (APP1099283 to D.L.W.); Cancer Council SA Beat Cancer Project on behalf of its donors and the State Government of South Australia through the Department of Health (MCF0418 to S.L.W., D.L.W.); Endeavour Research Fellowship from the Australian Government, Department of Education and Training (ERF_RDDH_179965 to J.N.); National Institute of Health (R01 to S.M.)

## Author contributions

J.N., T.J., D.L.W. and S.M. conceived and designed the study. J.N. and T.J. performed most of the experiments. C.L. developed methodology and performed blinded OA scoring. T.W. and A.A. analyzed scRNAseq dataset. Y.M., G.R., L.V., M.I., T.L., N.S., J.G. and H.K. assisted with animal husbandry, tissue collection and staining. S.W., D.J.H., T.C.W., D.R.H., D.M. and S.G. provided material or technical support, assisted with data analysis.

## Competing interests

The authors declare no competing interests.
