## [Peer Review File · Nature Communications]

Loss of Grem1-lineage chondrogenic progenitor cells causes osteoarthritis.Reviewer #1 (Remarks to the Author):

In this manuscript, Ng et al. study a population of Grem1-lineage articular cartilage cells, ultimately making the case that depletion of these cells is sufficient to drive osteoarthritis (OA) phenotypes. The findings are generally of interest, as the cellular pathogenesis of OA remains poorly understood and a clinically important question. While there are some requests for additional controls and clarification regarding the data, the underlying methodologies used are felt to be sound for most of the OA experiments. There are a few areas of concern, namely that the evidence presented here is not compelling that these cells are stem cells and there is some concern that the concepts of cell lineage versus component cell types are not discriminated as carefully as they should be in the writing. However, it is felt that these points can largely be addressed with text changes to make the claims of the manuscript better match the data presented. There are also some questions about results were the contributions or functions of growth plate resident versus articular cartilage Grem1-lineage cells may be conflated. Additional data is requested on that point. Also, there is a request for a little bit more though phenotyping of the OA models. If these issues can be addressed, there is enthusiasm for the findings reported here. Specific issues are described in more detail below.

1. The dual labeling with Grem creERT at the growth plate and articular cartilage after short chase periods suggests that these are two separate populations, or at least that the articular cartilage labeling does not necessarily indicate that these articular cartilage Grem1-lineage cells are derived from the growth plate Grem1-lineage cells. This conclusion is also supported by the scRNA-seq data. Given this, any other labeled populations observed, such as subchondral osteoblasts, could either be derived from growth plate resident or articular cartilage resident Grem 1+ cells. This also poses a challenge for the interpretation of the in vitro assays on FACS isolated cells in Fig 2, especially given the prior report from the authors on Grem1 cells. The current data presentation may give the impression that the labeled subchondral osteoblasts are derived from the articular cartilage cells. Similarly, the osteogenic potential of the articular cartilage Grem1 cells appears to be unclear and not strongly discriminated from that of the growth plate resident Grem1-lineage cells. In vitro assays may only reflect the function of the growth plate resident Grem1-lineage cells unless these two populations are separated and compared. Explicit clarification of this point regarding the interpretive challenges of distinguishing the roles of these two populations should be provided throughout the manuscript text and discussion. Some strategies for distinguishing the contribution of articular cartilage versus growth plate resident cells are also discussed below.

2. It would be helpful to define the relationship of Grem1-lineage articular cartilage cells to PRG4+ cells, as PRG4 has been a focus on study in a number of similar prior studies. It is also interesting and relevant to contrast the results here with those in a somewhat similar study on PRG4-lineage cells performed in Zhang et al. J Clin Invest 2016 in the discussion. The present work should also be compared to the overall similar results obtained with PRG4-creERT lineage tracing (e.g., Kozhemyakina et al. Arth and Rheum 2015; and work from the Chagin lab in Li et al. FASEB 2106 on this topic) and the phenotypes of PRG4-deficient mice in the discussion.

3. The data presented are insufficient to make the claim that Grem1-lineage articular cartilage cells represent stem cells. As these cells appear to be distinct from growth plate resident Grem1-lineage cells, it becomes important that all stemness assays separate and compare these populations, especially when the growth plate cells have already been nominated as stem cells in the authors' prior work. Moreover, the Chan et al. papers (Chan et al. Cell 2015 and Chan et al. Cell 2018) provide guidelines for key assays and standards needed to define skeletal stem cells, the core of which include transplantation of specific populations marked by combination of markers to demonstrate the stem-like properties of those cells in self-renewal and differentiation hierarchy studies. Notably, in vitro proliferation capacity as performed here has an uncertain relationship to stemness and is not strong evidence in support of identification of a stem cell population. Similarly, multilineage differentiation potential in vitro is a

widely distributed property in skeletal cells and not strong evidence for stemness. Also, reduction in a labeled population is not strong evidence of stemness, and the comparison to stem cell literature on this point should perhaps be tempered. However, it is appreciated that these studies would represent essentially a separate manuscript's worth of data. The present manuscript has interesting observations to communicate without this stem cell claim. Thus, it is recommended that all mentions of stem cells be removed from the text, section headings and figure legends and a more conservative term such as cartilage progenitor be used. It is appreciated that the investigators are likely sensitive to this issue through the use of the broad "CSP" term, but it is strongly recommended to go even further and only reserve nomination of this population for stemness as a possibility to be included in the discussion with both the strengths and limitations of this data considered.

4. Related to the point above, it is important to clarify in the text that the cells studied here using the Grem1-creERT are not an actual cell type, but rather a cell lineage, including one or more (likely a minimum of 2 given the early pulse-chase labeling in both growth plate and articular cartilage) initially labeled cell types and a number of derivatives of each of those initially labeled populations. Thus, discriminating specific cell types within the pool of Grem1-labeled cells would require additional staining of Grem1-creERT labeled cells with cell surface markers that would discriminate the initially labeled populations from the derivatives of these cells/determine the differentiation state of cells in the Grem1-lineage. It is appreciated that some relevant markers are examined in scRNA seq data, but these would instead have to be examined by FACS and in combination with cell isolation and transplantation studies of specific populations within the Grem1 lineage build a compelling case for distinguishing the different Grem1-lineage cell types. As with the above, it is felt that in light of the otherwise generally sufficient overall data package presented that it is not necessary to provide new experimental data to address this. Rather, modification of the language used to refer to these cells as Grem1-lineage cells and consistently clarify that lineages of cells and not specific cell types are being examined here would be sufficient.

5. Why was intra-articular diphtheria toxin administration used? Was there a GREM1+ population outside of bone that the investigators were trying to avoid targeting? Was there lethality observed with systemic diphtheria toxin administration? It would be helpful if this was added to the manuscript. When using the Grem1-creERT;DTR TdT mice, does the intra-articular injection only delete the articular cartilage or also the growth plate Grem1-lineage cells? Either result could make sense, but it would be helpful if this was spelled out.

6. Just to clarify, were the articular cartilage and growth plate separately isolated by microdissection from the Grem1-TdT mice and never subsequently pooled? Is this only the case for scRNA-seq experiments that these populations were separated via microdissection? This appears to be the case, but spelling it out a little bit more explicitly in the text and methods would be helpful for clarity.

7. It is implied in the data presentation and discussion that Foxo1 expression is selective for the articular cartilage and not growth plate resident Grem1-lineage cells. Is this correct? No data appears to be shown to address this point however. If this is correct, then this is an important and interesting finding that should be explored further to strengthen the manuscript. The presence and numbers of the Grem1-lineage cells at the articular cartilage versus growth plate sites in the foxo1 conditional knockout mice should be compared by histology and include quantitation to help support the selectivity of effect at the articular cartilage site. It would be ideal but perhaps not strictly necessary if flow cytometry on microdissected articular cartilage and growth plate specimens could also be used to support the selectivity of effect for only articular cartilage. Lastly, if these analyses do indeed support that only articular cartilage Grem1-lineage cells are strongly impacted by loss of Foxo1, then lineage tracing of the Grem1 cells in the resulting Grem1-creERT-TdT-Foxo1 fl/fl mice could be used to effectively only visualize the contribution of the growth plate resident but not articular cartilage resident Grem1 cells. This could help establish that these two pools of Grem1-lineage cells have distinct fates in vivo and thereby greatly strengthen the core claim of this work that there are distinct growth plate and articular cartilage resident Grem1-lineage cells.

8. The discussion is perhaps too cursory given the data presented. It is recommended that sections on the issue of there being at least two initially labeled Grem1-lineage cells that then need to be discriminated in downstream assays be given a thorough discussion. Other important issues to discuss are the question of whether or not these cells qualify as skeletal stem cells

9. The characterization of all of the OA models applied (especially as in Fig 4e, 4i and Fig 3j) should be improved to make a stronger case for the development of OA. In particular, images of the control groups need to be shown in Fig 3j. While it is appreciated that tol blue can respond to proteoglycan levels, it is also recommended to provide the widely utilized safranin O staining for comparison to other literature. Immunostaining with Col X would help to reinforce the hypertrophic phenotype of the chondrocytes observed. Staining for Prg4 should be considered as a means to show depletion of the superficial chondrocyte layer. As this is a particularly key point of the manuscript, additional images of the phenotype from different mice placed into extended data figs would be helpful. To this reviewer's eye, it is not so clear that the FGF18 treatment has much effect in the histology provided in Fig 4i. This image should be improved and other images from experimental replicates should be provided to give a more robust sense of the effects of treatment. It would also be helpful to look for classic OA-associated subchondral bone changes and osteophytes as evidence of OA effects using uCT. uCT in the diptheria toxin treatment model would also be helpful to distinguish the effects of DT treatment on articular cartilage versus growth plate resident Grem1-lineage cells, as presumably any changes in trabecular bone mass would likely reflect ablation of the growth plate resident but not articular cartilage populations.

10. Especially given that the FGF18 effects are hard to discern in the histology data provided, additional controls for FGF18 treatment will be important. At a minimum, it would be helpful to look for evidence of FGF18 signal transduction in the treated but not control group by staining for the activation downstream signaling mediators or known FGF target genes.

11. The scRNA-seq data is not presented in sufficient detail. First, some controls such as the number of unique genes or UMIs per cell should be displayed via pseudocoloring to assess for any artifacts related to sampling depth. Second, similar controls for mitochondrial and ribosomal RNA content should be similarly displayed by pseudocolor plots in sup figs. Scoring of proliferation markers, such as described in the Seurat documentation, is recommended to assess whether clustering is being driven by proliferation and whether such signals should be regressed out of the data. How were Each of the Chan et al. markers discussed (e.g., CD200, CD105, ect...) should be displayed in separate pseudocolor tSNE/U-MAP and violin plots. These should also be compared to classic cell type defining markers, including osteoblast markers (especially osteocalcin), and chondrocyte markers, including Sox9, Col2a1, aggrecan, COMP, growth plate markers including PTHrP. Some of this is mentioned in the methods text, but plots should be shown. How was it decided how many PCs to use for clustering? After isolating the different regions of Grem1/LEPR+ cells where these then pooled prior to sequencing or were these sequenced separately? If the latter, the different run batches should be compared. If the former, then how are the Grem-lineage cells discriminated from each other? Fig 4b is too small and a much larger and more readable version should be provided, at least in sup figs. Lastly a table should be provided including an annotation of each cluster for identity and including the key cell type defining genes used to make that annotation.

12. "Seurat" is misspelled in line 654.

13. The title is worded a little too ambitiously relative to the data shown, as it is too much of a stretch to claim that all OA including human OA is due to Grem1 cell loss from the data shown. Instead it is recommended that the title instead focus on how Grem1 cell depletion is sufficient to trigger OA or a similar more carefully title that matches the data shown.

Reviewer #2 (Remarks to the Author):

In this manuscript, the authors posit they have identified a stem cell population in the articular cartilage based on expression of Grem1. To support this, they present lineage tracing data (endogenous and following injury) and molecular analyses. They nominate several previously identified regulators of articular cartilage integrity such as Foxo1 and Fgf18 as being important to the biology of this Grem1+ cell.

Unfortunately, it is entirely unclear how the Grem1+ cells differ from bulk articular chondrocytes. As the authors themselves show, lineage tracing with Acan presents essentially the same results as compared to Grem1 tracing. The authors suggest that Grem1 cells are lost selectively, but without co-staining for chondrocyte markers this is completely subjective. In the image shown in Fig. 1, the entire articular surface seems to be missing in the Grem1 traced animal while Acan and Lepr traced seem in much better shape. The authors would need to repeat the entire study with a Col2-GFP transgene if they want to proceed with this type of methodology.

As this first premise to distinguish Grem1+ cells in any way from Acan+ chondrocytes is shaky, the rest of the paper falls succumbs to the weight of the house of cards. The experiments are well designed and the data presented well, but I would expect (and have seen) similar data for Acan lineage traced animals. For example, AC is lost with time, thus the number of Grem1+ cells decreases with time. Similarly, eliminating chondrocytes with a DTR reporter causes OA. In addition, deletion of Foxo1 in chondrocytes causes OA.

If Col2-GFP was used concurrently, the authors could then conduct scRNA-seq on cells co-expressing both markers or just Col2+ and try and show some sort of molecular difference. However, I imagine this would likely not show much of note due to Grem1 marking most if not all the AC. In any case, that kind of data would go a long way to answering the question of whether or not these cells are a distinct population.

Note to Reviewers

We are grateful to the reviewers for their comments and suggestions that have helped us substantially improve our manuscript (NCOMMS-21-49447). As indicated in our responses below, we have taken all of these into account in the revised version of our manuscript.

In the revised manuscript, we have highlighted major changes made from the original version using **red text**. Please note that figure numbers in our response to the reviewers denote the ones in the revised version of our manuscript unless otherwise specified. **Blue text is from reviewers**, our responses in black.

REVIEWER COMMENTS

Reviewer #1 (Remarks to the Author):

In this manuscript, Ng et al. study a population of Grem1-lineage articular cartilage cells, ultimately making the case that depletion of these cells is sufficient to drive osteoarthritis (OA) phenotypes. The findings are generally of interest, as the cellular pathogenesis of OA remains poorly understood and a clinically important question. While there are some requests for additional controls and clarification regarding the data, the underlying methodologies used are felt to be sound for most of the OA experiments. There are a few areas of concern, namely that the evidence presented here is not compelling that these cells are stem cells and there is some concern that the concepts of cell lineage versus component cell types are not discriminated as carefully as they should be in the writing. However, it is felt that these points can largely be addressed with text changes to make the claims of the manuscript better match the data presented.

We agree, please see our response to Comment 3 & 4, Reviewer 1 below.

There are also some questions about results where the contributions or functions of growth plate resident versus articular cartilage Grem1-lineage cells may be conflated. Additional data is requested on that point.

Please see our response to Comments 1, 6 & 7, Reviewer 1.

Also, there is a request for a little bit more thorough phenotyping of the OA models.

Please see our response to Comment 9, Reviewer 1.

If these issues can be addressed, there is enthusiasm for the findings reported here.

We thank the reviewer for their enthusiasm, constructive comments and suggestions. We have now addressed all comments below.

Specific issues are described in more detail below. 1. The dual labeling with *Grem1* creERT at the growth plate and articular cartilage after short chase periods suggests that these are two separate populations, or at least that the articular cartilage labeling does not necessarily indicate that these articular cartilage *Grem1*-lineage cells are derived from the growth plate *Grem1*-lineage cells. This conclusion is also supported by the scRNA-seq data.

Given this, any other labeled populations observed, such as subchondral osteoblasts, could either be derived from growth plate resident or articular cartilage resident *Grem1*⁺ cells.

While the origin (from GP or AC) of the *Grem1*-lineage subchondral bone population following post-natal induction is not the focus of our study, we agree that we do not provide data to support either origin. We do note that *Prg4*-lineage traced cells are not found in the GP, instead it is the superficial articular *Prg4*-lineage chondrocyte progenitors that give rise to subchondral bone (Li *et al* FASEB 2017). So it is likely that the articular *Grem1*-lineage similarly generates the subchondral bone, however we do not provide data to conclusively show this origin. We include this limitation of our study in the discussion as follows (line 310-315):

'A key difference between the *Prg4*- and *Grem1*-lineages in the tibio-femoral joint is that unlike *Prg4*-, *Grem1*- also gives rise to cells in the growth plate (Kozhemyakina *et al* 2015). Our focus has been on the articular *Grem1*-lineage population and superficial chondrocytes as this is the site of early OA-like changes. We cannot discount, however, that the GP *Grem1*-lineage CP cells may also contribute to the articular phenotypes observed and have not examined whether *Grem1*-subchondral bone traced populations arise from GP or AC progenitors or both.'

This also poses a challenge for the interpretation of the *in vitro* assays on FACS isolated cells in Fig 2, especially given the prior report from the authors on *Grem1* cells.

The current data presentation may give the impression that the labeled subchondral osteoblasts are derived from the articular cartilage cells. Similarly, the osteogenic potential of the articular cartilage *Grem1* cells appears to be unclear and not strongly discriminated from that of the growth plate resident *Grem1*-lineage cells. *In vitro* assays may only reflect the function of the growth plate resident *Grem1*-lineage cells unless these two populations are separated and compared.

Apologies, we have now clarified in the methods text (line 624-7) & on manuscript text (line 157) that the articular cartilage was first physically dissected prior to FACS-isolation of *Grem1*-lineage traced cells for use in the *in vitro* assays (**Fig 2**).

As such, the assays presented in **Fig 2** are indeed analysing the AC population, where our prior publication (Worthley *et al* 2015) isolated *Grem1*-lineage cells from whole bone preps (containing bone, marrow, GP and AC). The *Grem1*-lineage cells isolated from just the AC or whole bone had very similar properties upon culture *in vitro* with respect to

clonogenicity and multilineage differentiation potential to form osteogenic and chondrogenic, but not adipogenic, lineages (**Fig 2** and Worthley *et al* 2015). The separate GP and AC *Grem1*-lineage populations were micro-dissected prior to scRNAseq analysis and can be differentiated by their transcriptomes (**Fig 4**).

We have clarified this in the text:

Line 624-7 To limit contamination from non-articular cell populations present in whole bone preparations, the knee joints were dislocated at the epiphyses from both femur and tibia **to exclude the GP-cartilage**, gently disrupted using a mortar and pestle...

Line 157 These *Grem1*- and *Acan*-traced cells from the **AC of the** tibiofemoral joint were isolated via flow cytometry and plated at clonal density.

Explicit clarification of this point regarding the interpretive challenges of distinguishing the roles of these two populations should be provided throughout the manuscript text and discussion.

We acknowledge that there are two functionally distinct cartilaginous zones in the knee joint: the permanent articular cartilage and growth plate (line 74-77), consistent with the

two 'anatomically distinct' (line 105-6) *Grem1*-lineage populations. Aside from our scRNAseq data where we compare the *Grem1*-lineage GP and AC populations side by side, our focus is primarily on the AC *Grem1*-lineage cells as this is the site of early OA-like changes. However, we have now also quantified *Grem1*-lineage cell number in the growth plate following intra-articular treatment of *Grem1-DTR-TdTomato* mice and sacrifice 3 days later (as in **Fig.3g-o**). Different to the AC *Grem1*-lineage population, we find there is a trend towards decreased *Grem1*-lineage cells in the GP but no significant reduction in cell number compared to age matched control groups using this acute treatment regimen. We have added this information to the text and GP quantification to **Extended Data 3e** (shown

here) as follows:

Line 205-8 'This induced a significant reduction in *Grem1* CP cell number in the AC but not GP and increased blinded OA pathology scoring in DT treated *Grem1-DTR-TdTomato* mice compared to age-matched control groups (**Fig. 3h-o, Extended Data Fig 3e**).'

When we undertook blinded, qualitative histopathological analyses of growth plate morphology in these animals, understandably given the incomplete *Grem1*-lineage cell ablation in the GP using this dosing strategy, we detect only very mild changes to GP morphology in *Grem1-DTR-TdTomato* toxin treated animals in comparison to age and sex

matched controls. Further examination of the role of *Grem1*-lineage cells in the GP while interesting, would optimally involve ablation in younger mice than is relevant for early OA changes and is outside the scope of the current study. We cannot discount however that alterations to GP *Grem1*-lineage cells may also contribute to the articular phenotypes we observed across multiple transgenic models and with aging and so have added this qualification to the discussion text as follows:

Line 310-14 'A key difference between the *Prg4*- and *Grem1*-lineages in the tibio-femoral joint is that unlike *Prg4*-, *Grem1*- also gives rise to cells in the growth plate (Kozhemyakina *et al* 2015). Our focus has been on the articular *Grem1*-lineage population and superficial chondrocytes as this is the site of early OA-like changes. We cannot discount, however, that the GP *Grem1*-lineage CP cells may also contribute to the articular phenotypes observed...'

Some strategies for distinguishing the contribution of articular cartilage versus growth plate resident cells are also discussed below.

2. It would be helpful to define the relationship of *Grem1*-lineage articular cartilage cells to PRG4+ cells, as PRG4 has been a focus on study in a number of similar prior studies.

We agree and have now included an analysis to show the majority (63.7%) of AC cells express PRG4, 21.9% are labelled after short-term *Grem1*-lineage tracing and 12.6% are co-labelled by PRG4 and *Grem1*-lineage tracing, that is the two populations overlap but not completely. This is consistent with previous studies suggesting *Prg4* expression is restricted to the AC (Kozhemyakina *et al* 2015) and with our scRNAseq analysis showing significantly increased expression of *Prg4* in the *Grem1*-lineage AC, over GP cells (data now included in **Extended data 4g**).

We have included this information in the text and new data in **Extended data Fig 4**:

Line 235-41 'Consistent with previous studies, this included enrichment of the AC marker, *Lubricin* (*Prg4*), in AC compared to GP *Grem1*-lineage cells (Kozhemyakina *et al* 2015) (**Extended Data Fig.4f**). Immunofluorescence staining of tissue samples from two week lineage traced *Grem1*-*TdT* mice showed overlap between articular *Grem1*-lineage CP and *Prg4* expressing cells (12.6% of the articular cartilage double positive), but the *Grem1*-lineage cells were largely distinct from COL2-expressing chondrocytes in the calcified zone (<0.1% articular cartilage cells double positive for *Grem1*-lineage and COL2, **Extended Data Fig.4h**).'

It is also interesting and relevant to contrast the results here with those in a somewhat similar study on PRG4-lineage cells performed in Zhang *et al*. J Clin Invest 2016 in the discussion. The present work should also be compared to the overall similar results obtained with PRG4-creERT lineage tracing (e.g., Kozhemyakina *et al*. Arth and Rheum 2015; and work from the Chagin lab in Li *et al*. FASEB 2106 on this topic) and the phenotypes of PRG4-deficient mice in the discussion.

Our apologies for omitting this from the original manuscript. We have now added to the discussion:

Line 306-332 'The position of adult *Grem1*-lineage traced cells on the superficial surface of the AC, and subsequently in deeper layer chondrocytes and subchondral bone, is consistent with previous descriptions of chondrocytic label retaining progenitors and the *Prg4* (encoding lubricin) lineage where tracing began from birth or in juvenile mice, not adults as investigated here^{7,26}. A key difference between the *Prg4*- and *Grem1*-lineages in the tibio-femoral joint is that unlike *Prg4*-, *Grem1*- also gives rise to cells in the growth plate²⁶. Our focus has been on the articular *Grem1*-lineage population and superficial chondrocytes as this is the site of early OA-like changes. We cannot discount, however, that the GP *Grem1*-lineage CP cells may also contribute to the articular phenotypes observed and have not examined whether *Grem1*-lineage subchondral bone populations arise from GP or AC progenitors or both. We show that some, but not all, acutely labelled *Grem1*-lineage articular cells express *Prg4*/PRG4 and are distinct from the COL2-expressing chondrocytes of the calcified zone. Other key differences between the *Grem1*-lineage and that of the broader chondrocytic population marked by *Acan* include the extent and location of labelled lineage cells (**Fig. 2e, Extended Data Fig. 2b**), clonogenicity and differential potential *ex vivo* (**Fig. 2d-i**) and response to injury (**Fig. 1b-f**), such that *Grem1*- but not *Acan*-lineage cells were lost in the early OA-like phenotype induced by DMM surgery. Likewise, as FOXO1 expression is restricted to the superficial chondrocytes of the AC (**Extended Data Fig. 5b**), genetic deletion of *Foxo1* in *Grem1*-lineage cells effectively depletes FOXO1 expression in the articular surface, rather than deeper chondrocytic layers and results in OA. This suggests the *Grem1*-lineage is a specific subset of the total chondrocyte population that may be lost early in the disease process. We acknowledge that OA can be caused by dysfunction of mature cartilage cells, but here also highlight the important function of a specific chondrocyte progenitor population marked by *Grem1*. This is supported by phenotypic differences resulting from adult *Prg4* and *Grem1*-lineage ablation. Loss of both *Prg4*- and *Grem1*-lineage populations resulted in decreased superficial chondrocytes, however cartilage deterioration and OA histological features only occurred following loss of *Grem1*-, and not *Prg4*-, lineages (**Fig.3**) (Zhang *et al* 2016).'

3. The data presented are insufficient to make the claim that *Grem1*-lineage articular cartilage cells represent stem cells. As these cells appear to be distinct from growth plate resident *Grem1*-lineage cells, it becomes important that all stemness assays separate and compare these populations, especially when the growth plate cells have already been nominated as stem cells in the authors' prior work. Moreover, the Chan *et al.* papers (Chan *et al.* Cell 2015 and Chan *et al.* Cell 2018) provide guidelines for key assays and standards needed to define skeletal stem cells, the core of which include transplantation of specific populations marked by combination of markers to demonstrate the stem-like properties of those cells in self-renewal and differentiation hierarchy studies. Notably, *in vitro* proliferation capacity as performed here has an uncertain relationship to stemness and is not strong evidence in support of identification of a stem cell population. Similarly, multilineage differentiation potential *in vitro* is a widely distributed property in skeletal cells and not strong evidence for stemness. Also, reduction in a labeled population is not strong evidence of stemness, and the comparison to stem cell literature on this point

should perhaps be tempered. However, it is appreciated that these studies would represent essentially a separate manuscript's worth of data. The present manuscript has interesting observations to communicate without this stem cell claim. Thus, it is recommended that all mentions of stem cells be removed from the text, section headings and figure legends and a more conservative term such as cartilage progenitor be used. It is appreciated that the investigators are likely sensitive to this issue through the use of the broad "CSP" term, but it is strongly recommended to go even further and only reserve nomination of this population for stemness as a possibility to be included in the discussion with both the strengths and limitations of this data considered.

We agree with the reviewer that the stem cell point is not key to our findings and have added a short description of this limitation of our study to the discussion and amended the text of the manuscript to remove reference to stem and stem/progenitor cells and replaced with progenitor cells as follows:

Title Line 1 **Loss of *Grem1*-lineage chondrogenic progenitor cells is sufficient to cause osteoarthritis.**

Line 50 *Gremlin 1 (Grem1)* marks a novel **chondrogenic progenitor (CP)** cell

Line 52 Notably, this **CP** population is depleted when...

Line 54 ablation of *Grem1* **CP** cells in young mice

Line 58 caused proliferation of *Grem1*-lineage **CP** cells

Line 59-61 We propose that OA arises from the loss of **CP** cells at the articular surface secondary to an imbalance in **progenitor** cell homeostasis and present a new **progenitor** population as a locus for OA therapy.

Line 88-89 Here, we show that a unique population of ***Grem1*-lineage chondrogenic progenitor (CP)** cells,

Line 90-93 We next focused on the fate and function of these articular surface **CP** cells during aging, and upon OA-inducing injury. We find that during aging, and in two independent models of OA-inducing injury (DMM and CIOA), this **CP *Grem1*-lineage** is lost through apoptosis. Toxin-mediated ablation of these **CPs** in young mice also caused OA.

Line 96 FGFR3 agonists as potential therapeutic targets for *Grem1*-lineage **CP** cell maintenance and expansion.

Line 99 Injection of FGF18 into injured joints increased the number of *Grem1*-lineage **CP** cells...

Line 101 ...posits that OA is a disease caused by **CP** cell loss at the articular surface.

Line 104 *Grem1* **CP** cells are depleted in OA

Line 127 ...but rather is specific to the loss of the *Grem1*-lineage **CP** population.

Line 130 *Grem1*-TdT CIOA mice exhibited significant loss of *Grem1*-lineage **CP** cells through apoptosis...

Line 132 This suggested *Grem1*-lineage articular surface CP cells may be important in the pathophysiology of OA.

Line 134-42 ***Grem1* marks a chondrogenic progenitor population in the AC with osteoblastic lineage potential.** Given that OA models were characterised by loss of *Grem1*-lineage CP cells, we tested whether *Grem1*-lineage CP cells were a resident stem-progenitor cell for normal articular cartilage post-natal development and maintenance. *Grem1*-TdT and *Acan*-TdT mice were administered tamoxifen at postnatal day 4 to 6 before sacrifice with age matched *Lepr*-TdT mice, to determine the contribution of each lineage to the AC (Fig. 2a). *Grem1*-lineage CP cells were immediately observed within the cartilaginous epiphysis and meniscus after a week, and had given rise to 39.4% of the AC. In later stages of development, we found the *Grem1*-lineage CP...

Line 152 To further analyse the clonogenic and differentiation potential of *Grem1*-lineage CP cells...

Line 159 *Grem1*-lineage articular CP clones could be serially propagated...

Line 177 *Grem1*-lineage articular CP cells are lost with age

Line 180-4 *Grem1*-lineage articular CP cells from young to aged adult mice (Fig. 3b-c). This is consistent with other studies looking at the presence and regenerative capacity of skeletal stem cells in aged animals^{12,26}. The reduced regenerative capacity of the AC with age is due, at least in part, to reduced *Grem1*-lineage articular CP cells and chondrocyte proliferation (Extended Data Fig. 3a,b). We next examined whether *Grem1*-lineage CP cells persist in the aging knee...

Line 194 **Loss of *Grem1* articular CP cells causes OA.**

Line 200 ... investigate the consequences of adult *Grem1* CP cell loss in the AC...

Line 205-8 This induced a significant reduction in *Grem1* CP cell number in the AC but not GP and increased blinded OA pathology scoring in DT treated *Grem1*-DTR-TdTomato mice compared to age-matched control groups (Fig. 3h-o). To confirm the role for adult *Grem1* CP cells in OA...

Line 217-21...quantification of *Grem1* articular surface CP cells showed a significant loss in *Grem1* AC cells concomitant with worsened OA pathology, including a reduction in AC thickness (Extended Data Fig. 3f-g). Together our data confirmed that *Grem1* CP cells are normal progenitor cells

Line 226-7 ***Grem1* articular CP cells are distinct.** We applied scRNAseq analysis to characterise FACS-isolated early adult *Grem1*-lineage CP cells...

Line 233-42 ...clusters within the *Grem1*-lineage cells in the articular surface could also be separated by differential transcript expression from those in the GP... Most *Grem1*-lineage cells separately prepared from whole bone...

Line 245-6 This indicates that *Grem1*-lineage CP in the articular surface and mSSC in the GP partially overlap, but the articular surface *Grem1*-lineage population is distinct.

Line 250-56 We observed that *Foxo1* expression was highly correlated with *Grem1* expression ($p < 0.0001$) in articular CP cells in our scRNAseq data... To determine whether *Foxo1* expression was essential for survival of *Grem1* articular CP cells...

Line 260 Significantly fewer *Grem1*-lineage articular CP cells...

Line 263-65 This suggested that *Foxo1* was important to maintain adult *Grem1*-lineage articular CP cells and, by extension, AC integrity. Interestingly, *Grem1-TdT-Foxo1* mice administered tamoxifen in the early neonatal period (Day 4-6) developed drastic *Grem1*-lineage articular CP cell loss...

Line 268-9 This indicates the crucial role for *Foxo1* in AC development in the *Grem1*-lineage CP cell population, as well as in adult AC maintenance.

To discover molecular targets in *Grem1*-lineage articular CP cells for OA therapy...

Line 272-3 *Fibroblast growth factor receptor 3 (Fgfr3)* was expressed both in *Grem1*-lineage GP and articular CP populations.

Line 278-81 FGF18 treatment resulted in a significant increase in both the number of EdU+ *Grem1*-lineage articular CP cells and AC thickness, suggesting FGF18 induces AC chondrogenesis in the adult knee via increased *Grem1*-lineage articular CP cell proliferation.

Line 282-3 suggesting that the neo-cartilage arises from the articular CP population.

Line 285-6 FGF18 injection resulted in a significant increase in *Grem1*-lineage CP cells...

Line 287-8 This suggested that exogenous FGF18 alleviates OA through increased *Grem1*-lineage CP cell proliferation.

Line 300-2 In this study, we show that *Grem1* -lineage CP cells contribute to neonatal generation of the AC and are pivotal to AC maintenance in adulthood.

Line 334-41 Similar to mSSC cells¹², *Grem1*-lineage articular CP cells were depleted with age. Using two mouse models of OA, we found that *Grem1*-lineage CP cells in the knee joint were lost in disease. For the first time, we also report that functional ablation of *Grem1*-lineage articular CP cells, or genetic deletion of *Foxo1* in *Grem1*-lineage cells, in early adult mice led to significant and rapid OA. scRNAseq analysis identified separate populations of *Grem1*-lineage cells within the AC and GP structures, that provided an insight to the restricted commitment of each population of cells. FGF18 treatment induced proliferation of *Grem1*-lineage articular CP cells...

Line 343-331 We propose that the initial stage of OA can be predisposed by inadequate reserves of, or injury to, articular CP cells (accompanied by surface proteoglycan loss and fibrillation), followed by the apoptotic death of further CP cells, causing the inability to regenerate articular cartilage. This study reframes OA as a degenerative disease that can result from CP cell loss and provides a new focus for OA therapy.

Figure 1-3 legends:

Fig. 1 | The articular *Grem1*-lineage progenitor cells are significantly depleted in OA.

Fig. 2 | *Grem1*-lineage marks a progenitor cell population in the AC.

Fig. 3 | *Grem1*-lineage articular progenitors cells are lost with age and targeted ablation causes OA.

In extended data figure legends text '*Grem1*-lineage articular chondrogenic progenitor (CP) cells' now replaces '*Grem1* stem cells'. (**Ext data Fig 4**).

4. Related to the point above, it is important to clarify in the text that the cells studied here using the *Grem1*-creERT are not an actual cell type, but rather a cell lineage, including one or more (likely a minimum of 2 given the early pulse-chase labeling in both growth plate and articular cartilage) initially labeled cell types and a number of derivatives of each of those initially labeled populations. Thus, discriminating specific cell types within the pool of *Grem1*-labeled cells would require additional staining of *Grem1*-creERT labeled cells with cell surface markers that would discriminate the initially labeled populations from the derivatives of these cells/determine the differentiation state of cells in the *Grem1*-lineage. It is appreciated that some relevant markers are examined in scRNA seq data, but these would instead have to be examined by FACS and in combination with cell isolation and transplantation studies of specific populations within the *Grem1* lineage build a compelling case for distinguishing the different *Grem1*-lineage cell types. As with the above, it is felt that in light of the otherwise generally sufficient overall data package presented that it is not necessary to provide new experimental data to address this. Rather, modification of the language used to refer to these cells as *Grem1*-lineage cells and consistently clarify that lineages of cells and not specific cell types are being examined here would be sufficient.

We agree that our lineage tracing studies analysed the *Grem1*-lineage and have further amended the text of the manuscript to clarify this as follows:

Line 54-6 Transcriptomic analysis and functional modelling in mice revealed articular surface *Grem1*-lineage cells are dependent on *Foxo1*; ablation of *Foxo1* in *Grem1*-lineage cells also led to early OA.

Line 58 ...caused proliferation of *Grem1*-lineage CP cells (but not hypertrophic AC)...

Line 84 Our initial study¹⁵ focused on *Grem1*-lineage tissue-resident SSCs in the GP of mice.

Line 88-9 Here, we show that a unique population of *Grem1*-lineage chondrogenic progenitor (CP) cells,

Line 93 ...this CP *Grem1*-lineage is lost through apoptosis.

Line 95 ...revealed distinct molecular features of *Grem1*-lineage cells...

Line 96 FGFR3 agonists as potential therapeutic targets for *Grem1*-lineage CP cell maintenance and expansion.

Line 99 Injection of FGF18 into injured joints increased the number of *Grem1*-lineage CP cells...

Line 106 Focusing our studies on the knee joint of adult mice, we observed two anatomically distinct populations in the AC and GP marked by *Grem1*-lineage tracing.

Line 108 As such, we first determined if the articular *Grem1*-lineage cells were affected under OA disease conditions.

Line 123 ... cells at the site of proteoglycan loss showed a significant decrease in the *Grem1*-lineage population only (Fig. 1f).

Line 127 ...but rather is specific to the loss of the *Grem1*-lineage CP population.

Line 130 ...exhibited significant loss of *Grem1*-lineage CP cells through apoptosis...

Line 132 This suggested *Grem1*-lineage articular surface CP cells may be important in...

Line 136-8 Given that OA models were characterised by loss of *Grem1*-lineage CP cells, we tested whether *Grem1*-lineage CP cells were a resident stem-progenitor cell for normal articular cartilage post-natal development and maintenance.

Line 140 *Grem1*-lineage CP cells were...

Line 142 In later stages of development, we found the *Grem1*-lineage...

Line 152 To further analyse the clonogenic and differentiation potential of *Grem1*-lineage CSP cells in the early adult knee...

Line 156 These *Grem1*- and *Acan*-traced cells from

Line 159 *Grem1*-lineage articular CP clones could be serially propagated, while *Acan* clones

Line 163 Of the *Grem1*- and *Acan*-lineage clones that underwent >3 passages...

Line 165 ...both adult *Grem1*- and *Acan*-lineage clones gave rise to...

Line 169 A significantly greater percentage of *Grem1*-lineage clones were...

Line 172-5 Using long-term cell fate tracing of adult *Grem1*-lineage cells in vivo, we observed these cells contributing to progenitor populations in the superficial zone of the AC and with a significant increase in the percentage of *Grem1*-lineage cells within the calcified zone of the AC with age (Fig. 2j-k).

Line 177 *Grem1*-lineage articular CP cells are lost with age

Line 180-81 Quantification of fluorescence images showed a significant decrease in *Grem1*-lineage articular CP cells from young to aged adult mice

Line 183-5 The reduced regenerative capacity of the AC with age is due, at least in part, to reduced *Grem1*-lineage articular CP cells and chondrocyte proliferation (Extended Data Fig. 3a,b). We next examined whether *Grem1*-lineage CP cells persist in the aging knee by...

Line 188 *Grem1*-lineage cell number decreased significantly with increasing adult age...

Line 190 At this stage, *Grem1-lineage* cells were mainly observed in...

Line 214-20 DTR expression on *Grem1-lineage* cells was induced by tamoxifen administration at 4 weeks of age, followed by local ablation of *Grem1-lineage* AC cells by intra-articular injection of DT at 8-weeks-old for 2 weeks, before animals were sacrificed at 26 weeks (Extended Data Fig. 3e). Analysis of OA pathology and quantification of *Grem1-lineage* articular surface CP cells showed a significant loss in *Grem1-lineage* AC cells concomitant with worsened OA pathology, including a reduction in AC thickness (Extended Data Fig. 3f-g). Together our data confirmed that *Grem1-lineage* CP cells are...

Line 258-60 *Grem1-TdT-Foxo1* mice were administered tamoxifen at 6 weeks of age to induce *Foxo1* deletion in *Grem1-lineage* cells and sacrificed at 26 weeks (**Fig. 4d**). Significantly fewer *Grem1-lineage* articular CP cells...

Line 263-65 This suggested that *Foxo1* was important to maintain adult *Grem1-lineage* articular CP cells and, by extension, AC integrity. Interestingly, *Grem1-TdT-Foxo1* mice administered tamoxifen in the early neonatal period (Day 4-6) developed drastic *Grem1-lineage* articular CP cell loss...

Line 268-70 This indicates the crucial role for *Foxo1* in AC development in the *Grem1-lineage* CP cell population, as well as in adult AC maintenance.

To discover molecular targets in *Grem1-lineage* articular CP cells for OA therapy...

Line 272 *Fibroblast growth factor receptor 3 (Fgfr3)* was expressed both in *Grem1-lineage* GP and articular CP populations.

Line 275 ...impact of exogenous FGF18 on *Grem1-lineage* AC stem cells *in vivo*.

Line 278-81 FGF18 treatment resulted in a significant increase in both the number of EdU+ *Grem1-lineage* articular CP cells and AC thickness, suggesting FGF18 induces AC chondrogenesis in the adult knee via increased *Grem1-lineage* articular CP cell proliferation.

Line 285 FGF18 injection resulted in a significant increase in *Grem1-lineage* CP cells...

Line 287-8 This suggested that exogenous FGF18 prevents OA through increased *Grem1-lineage* CP cell proliferation.

Line 334-40 Similar to mSSC cells¹², *Grem1-lineage* articular CP cells were depleted with age. Using two mouse models of OA, we found that *Grem1-lineage* CP cells in the knee joint were lost in disease. For the first time, we also report that functional ablation of *Grem1-lineage* articular CP cells, or genetic deletion of *Foxo1* in *Grem1-lineage* cells, in early adult mice led to significant and rapid OA. scRNAseq analysis identified separate populations of *Grem1-lineage* cells within the AC and GP structures, that provided an insight to the restricted commitment of each population of cells. FGF18 treatment induced proliferation of *Grem1-lineage* articular CP cells...

Line 532 Ablation of *Grem1-lineage* cells in *Grem1-creERT;DTR* mice...

Line 580 ...TUNEL positive *Grem1-lineage* cells were quantified...

Figure 1-4 legends:

Fig. 1 | The articular *Grem1*-lineage progenitor cells are significantly depleted in OA.

Fig. 2 | *Grem1*-lineage marks a progenitor cell population in the AC.

Fig. 3 | *Grem1*-lineage articular progenitors cells are lost with age and targeted ablation causes OA.

Fig. 4 | *Grem1*-lineage single cell transcriptomics revealed a distinct population of articular chondrocytes that can be targeted for therapy.

In legend text '*Grem1*-lineage cells' now replaces '*Grem1* cells'.

In extended data figure legends text '*Grem1*-lineage cells' now replaces '*Grem1* cells'. (**Ext data Fig 3, 4**).

5. Why was intra-articular diphtheria toxin administration used? Was there a *GREM1*+ population outside of bone that the investigators were trying to avoid targeting? Was there lethality observed with systemic diphtheria toxin administration? It would be helpful if this was added to the manuscript.

Yes, *Grem1* also lineage traces cells in the brain and gut as reported in our previous publications (Worthley *et al* 2015, Ichinose *et al* 2021). Ablation of *Grem1* cells in the gut is problematic (our unpublished data and previously reported in a similar transgenic mouse) and is the reason behind intra-articular administration. This information can be found in the methods section:

Line 530-32 *Grem1-Td-DTR* mice administered with DT developed a fatal small intestinal phenotype at day 4 for homozygous mice and at day 7 for heterozygous mice after DT injection. A similar phenotype has been previously reported⁵.

When using the *Grem1*-creERT;DTR TdT mice, does the intra-articular injection only delete the articular cartilage or also the growth plate *Grem1*-lineage cells? Either result could make sense, but it would be helpful if this was spelled out.

As detailed above in response to Rev1 point 1-

We have now quantified *Grem1*-lineage cell number in the growth plate following intra-articular treatment of *Grem1-DTR-TdTomato* mice and sacrifice 3 days later (as in **Fig.3g-o**). Different to the AC *Grem1*-lineage population, we find there is a trend towards decreased *Grem1*-lineage cells in the GP but no significant reduction in cell number compared to age matched control groups using this acute treatment regimen. This is useful as this acute treatment allows us to analyse the separate contributions of the GP and AC *Grem1*-lineage populations to the OA-like phenotype at this time point. We have added this information to the text and GP quantification to **Extended Data 3e**.

Line 205-8 This induced a significant reduction in *Grem1* CP cell number in the AC but not GP and increased blinded OA pathology scoring in DT treated *Grem1-DTR-TdTomato* mice compared to age-matched control groups (**Fig. 3h-o, Extended Data Fig 3e,f**).

6. Just to clarify, were the articular cartilage and growth plate separately isolated by microdissection from the *Grem1*-TdT mice and never subsequently pooled? Is this only the case for scRNA-seq experiments that these populations were separated via microdissection? This appears to be the case, but spelling it out a little bit more explicitly in the text and methods would be helpful for clarity. Our apologies that there was insufficient explanation of the isolation and analysis of cell populations in our original manuscript. We have now clarified this in the text:

- *In vitro* clonogenicity and differentiation studies used dissected AC population only (GP population was discarded) –

Line 157 These *Grem1*- and *Acan*-traced cells from the AC of the tibiofemoral joint were isolated via flow cytometry...

Methods Line 624-27 To limit contamination from non-articular cell populations present in whole bone preparations, the knee joints were dislocated at the epiphyses from both femur and tibia to exclude the GP-cartilage, gently disrupted using a mortar and pestle...

Figure 2e shows region isolated – legend text: . e, Articular joints, outlined in yellow, were used to isolate red cells for *in vitro* assays using flow cytometry.

- FACS comparison of *Grem1*-lineage cells with mSSC markers

Whole bone preparations (including marrow, subchondral bone, AC, GP, cortical bone) were used for FACS isolation of *Grem1*-lineage traced cells for this analysis of mSSC markers (eg CD200) in the *Grem1*-lineage cells. This information can be found on line 241

'Most *Grem1*-lineage cells separately prepared from whole bone expressed the defining marker of mouse mSSC, CD200...'

We have added to Methods Line 664 'Cleaned bones (i.e. whole bone preparation) were gently disrupted using a mortar and pestle, minced, and digested in...'

Extended data Fig 4c legend 'Immunophenotyping of *Grem1*-lineage cells from total bone prep of 8-week-old mice using flow cytometry showed that...'

- scRNAseq studies used microdissected *Grem1*-lineage AC and GP populations that were kept separated through cell isolation and analyses.

Methods Line 677-691 'AC and GP from *Grem1*-TdT mice were separately excised under a dissection microscope and kept as separate tissue samples (AC and GP) through subsequent mincing using a scalpel and digestion in 2.5mg/mL collagenase type II (#CLS-2, Worthington). Cleaned *LepR*-TdT whole bones were gently disrupted using a mortar and pestle, minced, and digested in 2.5mg/mL collagenase type I (#CLS-1, Worthington). The following antibodies were

used: anti-CD45 (#103111, BioLegend), anti-Ter119 (#116211, BioLegend), anti-CD31 (#102509, BioLegend), and DAPI (#D9542, Sigma-Aldrich). *LepR* cells were sorted based on forward and side scatter, single cells, live cells (DAPI negative), trilineage negative expression (CD45-Ter119-CD31-) and TdTomato positive fluorescence. *Grem1* cells from the AC or GP were sorted separately based on forward and side scatter, single cells, live cells (DAPI negative), and TdTomato positive fluorescence, to generate separate *Grem1*-lineage populations derived from the AC or GP. Individual cells were collected into single well in a 96-well plate with lysis buffer previously prepared by the Sulzberger Genome Center (Columbia University) and sequenced. Cells were randomly allocated into 96-well plates to avoid batch effects.'

Extended Data Fig. 4 | a, Experimental schema showing tamoxifen dosing and analysis schedule and **b**, subsequent dissection of AC, GP or whole bone tissue from *Grem1-TdT* and *LepR-TdT* animals, with tissues from different sites or lineage tracing model kept separate as they were dissociated by digestion, separately subjected to FACS isolation of TdT positive stem/precursor cell populations, before scRNA sequencing in separate wells, then data from scRNA sequencing pooled for analysis.

7. It is implied in the data presentation and discussion that *Foxo1* expression is selective for the articular cartilage and not growth plate resident *Grem1*-lineage cells. Is this correct?

No data appears to be shown to address this point however. If this is correct, then this is an important and interesting finding that should be explored further to strengthen the manuscript. The presence and numbers of the *Grem1*-lineage cells at the articular cartilage versus growth plate sites in the *foxo1* conditional knockout mice should be compared by histology and include quantitation to help support the selectivity of effect at the articular cartilage site. It would be ideal but perhaps not strictly necessary if flow cytometry on microdissected articular cartilage and growth plate specimens could also be used to support the selectivity of effect for only articular cartilage.

Lastly, if these analyses do indeed support that only articular cartilage *Grem1*-lineage cells are strongly impacted by loss of *Foxo1*, then lineage tracing of the *Grem1* cells in the resulting *Grem1-creERT-TdT-Foxo1* fl/fl mice could be used to effectively only visualize the contribution of the growth plate resident but not articular cartilage resident *Grem1* cells. This could help establish that these two pools of *Grem1*-lineage cells have distinct fates in vivo and thereby greatly strengthen the core claim of this work that there are distinct growth plate and articular cartilage resident *Grem1*-lineage cells.

Thank you for this suggestion. Yes, *Foxo1* expression is significantly increased in *Grem1*-lineage cells of the articular cartilage compared to those in the growth plate. This can be observed in the Fig 4c violin plot and we have now also included an additional plot to **Extended data figure 5a** where we compare *Foxo1* expression across pooled AC or GP populations. While there is clear enrichment of *Foxo1* expression in the AC population, we can also detect *Foxo1*/FOXO1 expression in the GP. What is interesting is that the majority of FOXO1+ cells of the AC belong to the *Grem1*-lineage, but in comparison the FOXO1 and *Grem1*-lineage cell populations have very limited overlap in the GP. As suggested by

this reviewer, this provides some selectivity for *Foxo1* deletion using the *Grem1*-lineage cre mice for cells in the AC and not GP. This is also reflected in our new quantification of the number of *Grem1*-lineage cells in the GP of *Grem1-TdT-Foxo1* deletion mice compared to *Grem1-TdT* mice. In contrast to the AC (**Fig 4f**), we found no significant difference in *Grem1*-lineage cells in the GP. We now include analysis of the FOXO1+/*Grem1*-lineage marked cells in the AC and GP in **Extended data Fig. 5b**, quantification of *Grem1*-lineage cells in the GP of *Grem1-TdT-Foxo1* and control mice in **Extended data Fig. 5c** and have added this information to the text as follows:

Line 250-62 'We observed that *Foxo1* expression was highly correlated with *Grem1* expression ($p < 0.0001$) in articular CP cells in our scRNAseq data²⁹ (**Fig. 4c**), with *Foxo1* expression also significantly increased in the AC compared to GP *Grem1*-lineage populations (**Extended data Fig 5a**). At the protein level, the majority of FOXO1 expressing cells in the adult AC were *Grem1*-lineage cells, while there were very few FOXO1 expressing *Grem1*-lineage cells observed in the GP (**Extended data Fig. 5b**)... Significantly fewer *Grem1*-lineage AC but not GP cells were observed in *Grem1-TdT-Foxo1* mice, in comparison to age-matched *Grem1-TdT* controls, and the resulting increased OA pathology resembled *Grem1-creERT;DTR* DT ablation (**Fig. 4e-g, Extended data Fig 5c**). '

Acute ablation of *Grem1*-lineage cells in *Grem1-DTR-TdTomato* mice resulted in preferential reduction in the AC over the GP *Grem1*-population and resulted in the early OA-like phenotype. However we cannot exclude that the GP *Grem1*-population may play a role at later time points or with aging. Similarly *Foxo1* expression is enriched in the AC over the GP suggesting the AC *Grem1*-lineage/*Foxo1* expressing cells are likely to be key regulators of OA development. However we cannot be entirely conclusive on this point as there is still some *Foxo1*/FOXO1 expression in the GP *Grem1*-lineage. As the focus of our work has not been to investigate the GP *Grem1* population or the role of *Grem1*-lineage cells in the GP we now clarify these limitations to our study in the text (line 311-15).

8. The discussion is perhaps too cursory given the data presented. It is recommended that sections on the issue of there being at least two initially labeled *Grem1*-lineage cells that then need to be discriminated in downstream assays be given a through discussion.

Other important issues to discuss are the question of whether or not these cells qualify as skeletal stem cells

We have added a discussion of these points to the discussion text, line 302-05.

9. The characterization of all of the OA models applied (especially as in Fig 4e, 4i and Fig 3j) should be improved to make a stronger case for the development of OA. In particular, images of the control groups need to be shown in Fig 3j.

While it is appreciated that tol blue can respond to proteoglycan levels, it is also recommended to provide the widely utilized safranin O staining for comparison to other literature.

We have added Safranin O staining and additional control images to **Extended data Figs 3f, 5i**, plus provided an alternate higher quality image to **Fig 4i** tol blue.

Immunostaining with Col X would help to reinforce the hypertrophic phenotype of the chondrocytes observed. Staining for Prg4 should be considered as a means to show depletion of the superficial chondrocyte layer. As this is a particularly key point of the manuscript, additional images of the phenotype from different mice placed into extended data figs would be helpful.

We have added additional ColX immunofluorescent staining of hypertrophic chondrocytes and osteoarthritis phenotype to **Extended Data figs 3g, 4h**.

To this reviewer's eye, it is not so clear that the FGF18 treatment has much effect in the histology provided in Fig 4i. This image should be improved and other images from experimental replicates should be provided to give a more robust sense of the effects of treatment.

We agree the FGF18 treatment has a limited effect on the ColVII phenotype and can be variable between individual mice. We have added additional images from experimental replicates to **Extended data Fig 5i**.

It would also be helpful to look for classic OA-associated subchondral bone changes and osteophytes as evidence of OA effects using uCT. uCT in the diphtheria toxin treatment model would also be helpful to distinguish the effects of DT treatment on articular cartilage versus growth plate resident *Grem1*-lineage cells, as presumably any changes in trabecular bone mass would likely reflect ablation of the growth plate resident but not articular cartilage populations.

We have previously examined the role of *Grem1*-lineage cells in postnatal skeletogenesis using a diphtheria toxin ablation model (*Grem1*-creERT;R26-LSL-ZsGreen;R26-LSL-DTA), in which the *Grem1*-lineage ablation is not as efficient as the DTR models we have employed in this manuscript (Worthley *et al* 2015). Nevertheless, post-natal ablation of *Grem1*-lineage cells led to significantly reduced femoral bone volume by microCT after two weeks, with reduced trabecular bone as quantified by histology (Worthley *et al* 2015).

For clarity we have now added this information to the manuscript:

Line 195-9 'We have previously examined the role of *Grem1*-lineage cells in postnatal skeletogenesis using a diphtheria toxin ablation model (*Grem1*-creERT;R26-LSL-ZsGreen;R26-LSL-DTA), in which the *Grem1*-lineage was incompletely ablated (Worthley *et al* 2015). Nevertheless, post-natal ablation of *Grem1*-lineage cells led to significantly reduced femoral bone volume by microCT after two weeks, with reduced trabecular bone as quantified by histology (Worthley *et al* 2015).'

We did also score subchondral bone sclerosis and vascular invasion, osteocyte size and maturity as part of the OA pathology scoring matrix and have now added this information to the manuscript:

Methods line 587 'OA pathology in the *Grem1-TdT-DTR* experiments were scored blinded using scoring parameters of AC damage (0-11), proteoglycan loss (0-5), chondrocyte hypertrophy (0-3), subchondral bone sclerosis and vascular invasion (both 0-3), osteocyte size and maturity (both 0-3) and meniscus pathology (0-20) as described. The average scores of 2 – 3 slides per sample, total of 8 parameters, were summed to produce the final score. The average OA scores of both tibia and femora were included in this study.'

Line 208-11 'This increase in OA score with *Grem1*-cell ablation was predominantly due to cartilage hypertrophy, proteoglycan loss and structural damage, with a lesser contribution from meniscus pathology and subchondral vascular invasion and zero scoring from subchondral bone sclerosis or osteophyte changes.'

We respectfully suggest that OA effects on articular cartilage are fairly poorly measured by standard microCT analysis. It is clear from human studies that microCT effectively records changes in calcified tissue, but is poor for assessing cartilage degeneration. We suggest that uCT with phosphotungstic acid contrast, as used previously (Li *et al* 2017) to visualise soft tissues such as cartilage in the mouse, is outside the scope of the present study.

10. Especially given that the FGF18 effects are hard to discern in the histology data provided, additional controls for FGF18 treatment will be important. At a minimum, it would be helpful to look for evidence of FGF18 signal transduction in the treated but not control group by staining for the activation downstream signaling mediators or known FGF target genes.

We are grateful for the reviewer's suggestion and agree that it would be intriguing to examine the downstream molecular alterations resulting from FGF18 treatment. However, as the transcriptional outputs downstream of FGF signalling are context and time dependent we feel that an elucidation of which are the key players in the setting of FGF18 treatment of the articular cartilage in our OA model is a large body of work in itself and outside the scope of the current study.

11. The scRNA-seq data is not presented in sufficient detail.

We have now expanded the scRNA-seq analyses presented in the revised manuscript to address these concerns.

First, some controls such as the number of unique genes or UMIs per cell should be displayed via pseudocoloring to assess for any artifacts related to sampling depth.

Second, similar controls for mitochondrial and ribosomal RNA content should be similarly displayed by pseudocolor plots in sup figs.

The majority of cells have unique molecular identifiers (UMI) of >1000, consistent with high quality scRNAseq data sequenced to sufficient depth in our analyses. We have now referred to this analysis in the methods text (line 699-02) and added pseudocoloured tSNE plots to display UMI, mitochondrial and ribosomal RNA content per cell to **extended data figure 4c, d**.

Line 699-02 We undertook quality control assessments to check that sequencing depth (indicated by unique molecular identifiers per cell), sample quality (mitochondrial and ribosomal content per cell) and proliferation markers were not drivers of cluster identity (**Extended Data Fig. 4d**).

Scoring of proliferation markers, such as described in the Seurat documentation, is recommended to assess whether clustering is being driven by proliferation and whether such signals should be regressed out of the data.

We have assessed proliferation markers (S phase: *Mcm5, Pcna, Tyms, Fen1, Mcm2, Mcm4, Rrm1, Gins2, Mcm6, Cdca7, Dtl, Prim1, Hells, Rfc2, Rpa2, Nasp, Gmnn, Wdr76, Slbp, Ccne2, Pold3, Msh2, Atad2, Rad51, Rrm2, Cdc45, Cdc6, Exo1, Tipin, Dscc1, Blm, Casp8ap2, Clspn, Pola1, Chaf1b, Brip1*. G2M phase: *Hmgb2, Cdk1, Nusap1, Birc5, Tpx2, Top2a, Ndc80, Cks2, Nuf2, Cks1b, Mki67, Tmpo, Cenpf, Tacc3, Smc4, Ccnb2, Ckap2l, Ckap2, Aurkb, Kif11, Anp32e, Tubb4b, Kif20b, Hjurp, Cdca3, Hn1, Cdc20, Kif2c, Rangap1, Ncapd2, Dlgap5, Cdca2, Cdca8, Ect2, Kif23, Hmnr, Aurka, Psrc1, Anln, Lbr, Ckap5, Cenpe, Cctf, Nek2, G2e3, Gas2l3, Cbx5, Cenpa*) in the dataset. We have regressed these proliferation markers from the dataset and can confirm it does not alter our analysis-PCA is unchanged before and after regression of cell cycle marker expression. PCA plots included below for the reviewer but not included in the revised manuscript, however information that we have undertaken this additional quality control metric is now included in methods section line 701.

Each of the Chan et al. markers discussed (e.g., CD200, CD105, etc...) should be displayed in separate pseudocolor tSNE/U-MAP and violin plots. These should also be compared to classic cell type defining markers, including osteoblast markers (especially osteocalcin), and chondrocyte markers, including Sox9, Col2a1, aggrecan, COMP, growth plate markers including PTHrP. Some of this is mentioned in the methods text, but plots should be shown.

We have added tSNE and violin plots to **extended data figure 4f**.

How was it decided how many PCs to use for clustering?

We used a dimension reduction setting from 1 to 9 to capture the majority of the variation in the data, as visualised by the point of inflection of the 'elbow' plot. We have now added this information to the methods section and the elbow plot to **extended data figure 4c**.

Line 696-98 When using FindClusters() function, dimensions of reduction were set from 1 to 9 and resolution was set at 0.6 to capture the majority of variation in the data (**Extended Data Fig. 4c**).

After isolating the different regions of *Grem1*/LEPR+ cells were these then pooled prior to sequencing or were these sequenced separately? If the latter, the different run batches should be compared. If the former, then how are the *Grem1*-lineage cells discriminated from each other?

The *Grem1*-lineage cells were physically dissected from the AC or GP from the same animals and then maintained as separate populations and barcoded as such, but sequenced on the same run. We have clarified this with additional methods and extended data figure legend details as follows:

Methods Line 677-91 AC and GP from *Grem1-TdT* mice were **separately** excised under a dissection microscope and **kept as separate tissue samples (AC and GP) through subsequent** mincing using a scalpel and digestion in 2.5mg/mL collagenase type II (#CLS-2, Worthington). Cleaned *LepR-TdT* whole bones were gently disrupted using a mortar and pestle, minced, and digested in 2.5mg/mL collagenase type I (#CLS-1, Worthington). The following antibodies were used: anti-CD45 (#103111, BioLegend), anti-Ter119 (#116211, BioLegend), anti-CD31 (#102509, BioLegend), and DAPI (#D9542, Sigma-Aldrich). *LepR* cells were sorted based on forward and side scatter, single cells, live cells (DAPI negative), trilineage negative expression (CD45-Ter119-CD31-) and TdTomato positive fluorescence. *Grem1* cells **from the AC or GP** were sorted **separately** based on forward and side scatter, single cells, live cells (DAPI negative), and TdTomato positive fluorescence, **to generate separate *Grem1*-lineage populations derived from the AC or GP**. Individual cells were collected into single well in a 96-well plate with lysis buffer previously prepared by the Sulzberger Genome Center (Columbia University) and sequenced. Cells were randomly allocated into 96-well plates to avoid batch effects.

Extended Data Fig. 4 | a, Experimental schema showing tamoxifen dosing and analysis schedule and **b**, subsequent dissection of AC, GP or whole bone tissue from *Grem1-TdT* and *LepR-TdT* animals, **with tissues from different sites or lineage tracing model kept separate** as they were dissociated by digestion, **separately** subjected to FACS isolation of TdTomato positive

progenitor cell populations, before scRNA sequencing **in separate wells**, then data from scRNA sequencing pooled for analysis.

Fig 4b is too small and a much larger and more readable version should be provided, at least in sup figs.

We have added a larger version to **extended data figure 4e**. To enable easy readability, differentially expressed transcripts across the cluster are also listed in **extended data table 1**.

Lastly a table should be provided including an annotation of each cluster for identity and including the key cell type defining genes used to make that annotation.

We have added this to **extended data table 1**.

12. "Seurat" is misspelled in line 654.

This has been corrected.

13. The title is worded a little too ambitiously relative to the data shown, as it is too much of a stretch to claim that all OA including human OA is due to Grem1 cell loss from the data shown. Instead it is recommended that the title instead focus on how Grem1 cell depletion is sufficient to trigger OA or a similar more carefully title that matches the data shown.

We have modified the manuscript title to '**Loss of Grem1-lineage chondrogenic progenitor cells is sufficient to cause osteoarthritis**'.

Reviewer #2 (Remarks to the Author):

In this manuscript, the authors posit they have identified a stem cell population in the articular cartilage based on expression of Grem1. To support this, they present lineage tracing data (endogenous and following injury) and molecular analyses. They nominate several previously identified regulators of articular cartilage integrity such as Foxo1 and Fgf18 as being important to the biology of this Grem1+ cell.

Unfortunately, it is entirely unclear how the Grem1+ cells differ from bulk articular chondrocytes. As the authors themselves show, lineage tracing with Acan presents essentially the same results as compared to Grem1 tracing. The authors suggest that Grem1 cells are lost selectively, but without co-staining for chondrocyte markers this is completely subjective. In the image shown in Fig. 1, the entire articular surface seems to be missing in the Grem1 traced animal while Acan and Lepr traced seem in much better shape. The authors would need to repeat the entire study with a Col2-GFP transgene if they want to proceed with this type of methodology.

As this first premise to distinguish *Grem1*⁺ cells in any way from *Acan*⁺ chondrocytes is shaky, the rest of the paper falls succumbs to the weight of the house of cards. The experiments are well designed and the data presented well, but I would expect (and have seen) similar data for *Acan* lineage traced animals.

For example, AC is lost with time, thus the number of *Grem1*⁺ cells decreases with time.

Similarly, eliminating chondrocytes with a DTR reporter causes OA. In addition, deletion of *Foxo1* in chondrocytes causes OA.

If *Col2*-GFP was used concurrently, the authors could then conduct scRNA-seq on cells co-expressing both markers or just *Col2*⁺ and try and show some sort of molecular difference. However, I imagine this would likely not show much of note due to *Grem1* marking most if not all the AC. In any case, that kind of data would go a long way to answering the question of whether or not these cells are a distinct population.

Firstly, we agree with the reviewer that the disruption of chondrocytes is certainly an etiology for OA. We do not dispute this claim in this paper and have altered the text to make this more clear. However, we do claim that the death of a distinct population of cartilage progenitors, found in the articular surface and distinct in gene expression, location and functional progenitor capacity is *also* responsible for early OA. We argue that there are clear and important differences that separate the *Grem1*-lineage from the broader chondrocytic population, where *Grem1*-lineage loss can be an early event that is critical to understanding OA development. This population responds to FGF18 and can be partially rescued by FGF18 injection. We feel these findings are novel and merit publication.

We appreciate the reviewer's sentiment and agree that it is important to address what appears to be their key concern, i.e. to clarify and expand upon our discussion of the differences between *Grem1*-lineage traced cells and the broader chondrocyte populations marked by *Acan* or *Col2*/COL2.

We agree that there are similarities between the *Grem1*- and *Acan*-lineages, indeed the *Acan*-transgenic mouse line was included in the study to mark the broad pool of chondrocytes present in the knee joint and to allow an assessment of the effect of aging and early OA phenotype induced by DMM surgery on this broad chondrocyte population. We know from previous studies using the *Acan* homozygous hypomorph mouse (*Acan*-creERT2 allele) that decreased *Acan* expression results in a postnatal dwarf phenotype and severe cartilage erosion of the knee joint at 12months of age-highlighting the key role for this gene in cartilage function.

However we would respectfully suggest that there are also important key differences between the *Grem1*-lineage and that of the broader *Acan*-or *Col2*/COL2 marked populations, namely:

1. marker expression and location
2. clonogenic capacity ex vivo
3. differentiation potential
4. response to aging and injury.

We discuss each point more fully below.

1. Marker expression and location

Firstly, it is not unusual in the field of stem cell biology that expression of marker genes can overlap between progenitor and progeny populations, with specific markers becoming more restricted as the population becomes more differentiated. Similar to this we observe that the *Grem1*- and *Acan*- lineages overlap, but we also observe differences as described in the manuscript text:

(line 154-56 *in vivo* lineage tracing) '*Grem1* labelled specific articular surface cells (**Fig. 2e**) overlapping with, but more limited than, the total *Acan* chondrocyte population (**Extended Data Fig. 2b**).'

We have improved the labelling of **Extended Data Figure 2b** to highlight differences in cell populations observed in *Acan*-, compared to *Grem1*-lineage traced knee joints, i.e. *Acan*, but not *Grem1*-, lineage traces the majority of the meniscus of the knee joint and anterior cruciate ligament also (**Fig 2b, Extended Data Figure 2b**)

The *Grem1*-lineage also traces cells outside of those marked by *Acan* and COL2 (e.g. non-cartilage populations in the brain and gut mesenchyme –see Worthley *et al* 2015, Ichinose *et al* 2021). Together this suggests that these important lineage markers are differentially regulated *in vivo* and while generating some overlapping chondrocytic cell populations the lineage populations are also distinct. We speculate they may also have capacity to respond to different environmental and spatiotemporally regulated cues.

We now also include co-immunofluorescence staining of the articular cartilage of the knee joint to show that short-term *Grem1*-lineage traced cells are a discrete population of progenitor cells to those deeper layer chondrocytes that express COL2 (**Extended data fig 4h**). We believe this addresses the reviewers concerns about whether or not the *Grem1*-lineage and COL2-expressing cells of the articular cartilage are appreciably different.

2. Clonogenic capacity *ex vivo*

This is described in the manuscript text:

(line 159-63 *ex vivo* culture and expansion/self-renewal properties) '*Grem1*-lineage articular CP clones could be serially propagated, while *Acan* clones lost serial propagation (**Fig. 2f**). *Grem1* expression was also significantly higher in *Acan* clones capable of *in vitro* expansion compared to those that were not (**Extended Data Fig. 2d**), suggesting *Grem1* expression correlated with self-renewal *in vitro*.'

3. Differentiation potential

We observe differences in the *ex vivo* osteogenic potential of *Grem1*- and *Acan*-lineage clones as described in the manuscript text:

(line 168-70) 'A significantly greater percentage of *Grem1*-lineage clones were capable of osteogenic differentiation compared to *Acan*-lineage clones (100% vs 25%) (**Fig. 2i**).

4. Response to aging and injury

Acan transcript expression is maintained in the spontaneous mouse model of OA, STR/ort, and in old mice (Chambers *et al.* 2002). In contrast, we observe that *Grem1*-lineage cells are lost with aging in our model. Yes, the AC is generally lost in older age but we show that even when

normalised to the total number of chondrocytes present in aged knee joints, the proportion of *Grem1*⁺ cells is significantly decreased (**Fig 3a**). We aspire to explain the reason for that through an examination of early changes to the underlying progenitor cell populations. Our data also shows that *Grem1*⁻, but not *Acan*⁻, lineage cells are lost in the early OA-like phenotype induced by DMM-surgery (**Fig 1e**). This suggests the *Grem1*-lineage, but not the entire chondrocytic population, may be lost early in the disease process.

We describe this notion in the text-line 125-8:

'Notably, the persistence of *Acan* AC chondrocytes in DMM animals suggests that the initial stage of OA is not due to total chondrocyte loss, but rather is specific to the loss of the *Grem1*-lineage CP population.'

5. Evidence that the *Grem1*-lineage specifically is important in the OA phenotype.

Administration of diphtheria toxin to our *Grem1-DTR-Td* mouse, resulted in ablation of *Grem1*-expressing AC cells. Loss of these *Grem1*-expressing cells alone, rather than ablation of the entire chondrocytic population, is sufficient to generate OA. We now include new staining data to show that this acute toxin treatment in these mice also resulted in upregulation of the hypertrophic COLX marker in the AC, in comparison to toxin treated controls. While we acknowledge that broad chondrocyte dysfunction can generate OA, this data shows that loss of the *Grem1*-cells using an acute toxin treatment regimen is enough to generate OA.

We have also now performed immunofluorescence staining against FOXO1 and show that the FOXO1 positive cells are primarily restricted to the articular surface (consistent with a previous report, Lee *et al.*, 2020) - and co-stain with the *Grem1*-lineage marker, but not with cells in the deeper chondrocytic layers of the AC (**Extended data fig 5b**).

Inducible deletion of *Foxo1* in *Grem1*-expressing cells (using the *Grem1-creERT;Foxo1-fl* mice) therefore only effectively depletes FOXO1 expression in this articular surface since FOXO1 is not even expressed in the deeper chondrocytes of the AC (again highlighting the unique gene expression pattern found in *Grem1*-lineage cells). Therefore the OA phenotype we observe following *Foxo1* ablation in *Grem1*-expressing cells results from this alteration in the superficial chondrocytic progenitors rather than deeper chondrocytic layer cells. Highlighting once again, that perturbations to the *Grem1*-lineage cells can result in OA.

As such, we respectfully include a discussion of these points in our revised manuscript as follows (line 291-332):

'OA, like other degenerative diseases, can be viewed **as the dysfunction of mature cartilage cells or the result of an imbalance in stem-progenitor cell repair and renewal**. Previous studies have shown functional cartilage cell dysfunction/death results in OA, here we show that **ablation of a restricted progenitor population can cause OA without disturbing the broader population of differentiated chondrocytes...**

...The position of adult *Grem1*-lineage traced cells on the superficial surface of the AC, and subsequently in deeper layer chondrocytes and subchondral bone, is consistent with previous descriptions of chondrocytic label retaining progenitors and the *Prg4* (encoding lubricin) lineage where tracing began from birth or in juvenile mice, not adults as investigated here (Kozhemyakina *et al* 2015, Li *et al* 2017). A key difference between the *Prg4*⁻ and *Grem1*⁻ lineages in the tibio-femoral joint is that unlike *Prg4*⁻, *Grem1*⁻ also gives rise to cells in the

growth plate (Kozhemyakina *et al* 2015). Our focus has been on the articular *Grem1*-lineage population and superficial chondrocytes as this is the site of early OA-like changes. We cannot discount, however, that the GP *Grem1*-lineage CP cells may also contribute to the articular phenotypes observed and have not examined whether *Grem1*-lineage subchondral bone populations arise from GP or AC progenitors or both. We show that some, but not all, acutely labelled *Grem1*-lineage articular cells express *Prg4*/PRG4 and are distinct from the COL2-expressing chondrocytes of the calcified zone. Other key differences between the *Grem1*-lineage and that of the broader chondrocytic population marked by *Acan* include the extent and location of labelled lineage cells (**Fig. 2e, Extended Data Fig. 2b**), clonogenicity and differential potential *ex vivo* (**Fig. 2d-i**) and response to injury (**Fig. 1b-f**), such that *Grem1*- but not *Acan*-lineage cells were lost in the early OA-like phenotype induced by DMM surgery. Likewise, as FOXO1 expression is restricted to the superficial chondrocytes of the AC (**Extended Data Fig. 5b**), genetic deletion of *Foxo1* in *Grem1*-lineage cells effectively depletes FOXO1 expression in the articular surface, rather than deeper chondrocytic layers and results in OA. This suggests the *Grem1*-lineage is a specific subset of the total chondrocyte population that may be lost early in the disease process. This is supported by phenotypic differences resulting from adult *Prg4* and *Grem1*-lineage ablation. Loss of both *Prg4*- and *Grem1*-lineage populations resulted in decreased superficial chondrocytes, however cartilage deterioration and OA histological features only occurred following loss of *Grem1*-, and not *Prg4*-, lineages (**Fig.3**, Zhang *et al* 2016).

Rebuttal references

- Chambers, M.G. *et al.* Expression of collagen and aggrecan genes in normal and osteoarthritic murine knee joints. *Osteoarthritis and Cartilage* **10**, 51-61, (2002).
- Chan, C. K. *et al.* Identification and specification of the mouse skeletal stem cell. *Cell* **160**, 285-298, (2015).
- Ichinose, M. *et al.* The BMP antagonist gremlin 1 contributes to the development of cortical excitatory neurons, motor balance and fear responses. *Development* **148**, doi:10.1242/dev.195883 (2021).
- Kozhemyakina, E. *et al.* Identification of a *Prg4*-expressing articular cartilage progenitor cell population in mice. *Arthritis Rheumatol* **67**, 1261-1273 (2015).
- Li, L. *et al.* Superficial cells are self-renewing chondrocyte progenitors, which form the articular cartilage in juvenile mice. *FASEB J* **31**, 1067-1084 (2017).
- Lee, K. *et al.* FOXO1 and FOXO3 transcription factors have unique functions in meniscus development and homeostasis during aging and osteoarthritis. *PNAS* **117** 3135-3143 (2020).
- Worthley, D. L. *et al.* Gremlin 1 identifies a skeletal stem cell with bone, cartilage, and reticular stromal potential. *Cell* **160**, 269-284, (2015).
- Zhang, M. *et al.* Induced superficial chondrocyte death reduces catabolic cartilage damage in murine posttraumatic osteoarthritis. *J Clin Invest.* **126**, 2893-2902, (2016).

Reviewer #1 (Remarks to the Author):

The authors have provided a satisfactory response to the points raised in the initial round of revisions.

One additional point is that to this reviewer's knowledge it is unlikely that meniscus labeling with Grem1-lineage reporters indicates that the meniscus or anterior cruciate ligament is derived from the Grem1+chondroprogenitor. It is more likely that the Grem1 promoter is independently active in each of these sites. This should be clarified in the text.

This reviewer fundamentally agrees with concerns that there remains significant uncertainty about the degree to which Grem1 cells truly represent a distinct population of chondrocytes and uncertainty about the degree to which the phenotypes observed reflect the specific function of a distinct population of Grem1-lineage cells and not generic effects of targeting chondrocytes. In particular, some of the responses offered on this point are not compelling. For instance as mentioned above, labeling of Grem1 in ligaments or in extra-skeletal tissues does not speak to whether Grem1 CPs are distinct from generic articular chondrocytes. Similarly, *in vitro* differentiation and CFU assays have no compellingly consistent relationship with *in vivo* skeletal biology and are probably best considered minor supporting evidence.

However, despite these shortcomings of there being remaining questions about the degree to which the Grem1 CP is distinct, the findings here remain of interest to this reviewer as long as there is careful qualification around this point. Additionally, scRNA-seq studies and co-staining immunofluorescence studies as in Ext Data Fig 4, while not definitively addressing this concern, at least provide initial consideration of the degree to which Grem1 marks a distinct population of chondrocytes.

Reviewer #2 (Remarks to the Author):

The effort put in by the authors to address the Reviewers' comments is appreciated. However, the primary issue of distinguishing Grem1+ cells from bulk articular chondrocytes remains unaddressed. This would be a very simple experiment that would build on the scRNA-seq data already presented: comparing sorted Acan-Cre traced cells with Grem1-Cre traced cells. If Grem1+ cells represent or contain a distinct population of cells, this experiment would clearly demonstrate this and potentially identify regulators of this difference. Given the strength of the claims made by the authors, i.e. that Grem1+ cells represent chondrocyte progenitors that are also involved in OA pathogenesis, defining these cells as functionally and molecularly distinct is of great importance and fundamental to the paper.

Without consistent use of Acan-Cre throughout the paper and comparing the results obtained with Grem1-Cre, the claim that Grem1 marks a chondrogenic progenitor cell is not distinguishable from the claim that Acan marks a chondrogenic progenitor cell. The use of Grem1 as a marker of CPs is the fundamental premise of the paper and the data in their current form do not support this. scRNA-seq would be the simplest way of addressing this, but until some molecular evidence to support the difference between these two populations is presented the paper does not provide much in the way of significance given the known roles of Foxo1 and Fgg118 in articular cartilage. *In vitro* assays and immunostaining are supportive of the claims, but functional (*in vivo*) and molecular comparisons between Grem1- and Acan-Cre cells are lacking and sorely needed to support the claims.

Note to Reviewers

We are grateful to the reviewers for their comments and suggestions that have helped us substantially improve our manuscript (NCOMMS-21-49447A), importantly now including key data showing *Grem1* marks a lineage that overlaps with *Acan/ACAN+* articular cartilage cells in the knee joint but is also distinct. As indicated in our responses below, we have included this information in the revised version of our manuscript.

In the revised manuscript, we have highlighted major changes made from the second submission using **red text**. Please note that figure numbers in our response to the reviewers denote the ones in the revised version of our manuscript unless otherwise specified. **Blue text is from reviewers**, our responses in black.

REVIEWER COMMENTS

Reviewer #1 (Remarks to the Author):

1. The authors have provided a satisfactory response to the points raised in the initial round of revisions.

One additional point is that to this reviewer's knowledge it is unlikely that meniscus labeling with *Grem1*-lineage reporters indicates that the meniscus or anterior cruciate ligament is derived from the *Grem1*+chondroprogenitor. It is more likely that the *Grem1* promoter is independently active in each of these sites. This should be clarified in the text.

With respect, labelling of the anterior cruciate ligament predominantly occurs in the *Acan*-lineage traced animals, not the *Grem1*-lineage, and is a key difference we observe between the locations of both populations of traced cells. This can be viewed in **Extended data Fig 2b**. We understand your point though and have addressed this in the discussion:

Line 366-68 **Equally we have not investigated which *Grem1*-lineage population gives rise to the meniscus or if the *Grem1* promoter is independently active at sites throughout the knee joint.**

2. This reviewer fundamentally agrees with concerns that there remains significant uncertainty about the degree to which *Grem1* cells truly represent a distinct population of chondrocytes and uncertainty about the degree to which the phenotypes observed reflect the specific function of a distinct population of *Grem1*-lineage cells and not generic effects of targeting chondrocytes. In particular, some of the responses offered on this point are not compelling. For instance as mentioned above, labeling of *Grem1* in ligaments or in extra-skeletal tissues does not speak to whether *Grem1* CPs are distinct from generic articular chondrocytes. Similarly, *in vitro* differentiation and CFU assays have no compellingly consistent relationship with *in vivo* skeletal biology and are probably best considered minor supporting evidence.

However, despite these shortcomings of there being remaining questions about the degree to which the *Grem1* CP is distinct, the findings here remain of interest to this reviewer as long as there is careful qualification around this point. Additionally, scRNA-seq studies and co-staining immunofluorescence studies as in Ext Data Fig 4, while not definitively addressing

this concern, at least provide initial consideration of the degree to which *Grem1* marks a distinct population of chondrocytes.

We thank the reviewer for their comments. To address this we now include a new scRNAseq dataset derived from short-term *Grem1*-lineage traced articular cells from the early adult knee (new **Fig. 3m**). This new analysis highlights that while many *Grem1* lineage cells express *Acan*, we were also able to identify *Acan*-only expressing and *Grem1*-only expressing populations. We have also undertaken scRNAseq analysis of *Acan*-lineage cells using an identical induction and collection protocol and integrated the two datasets as shown below (Review Figure 1) which again illustrates the overlapping transcriptional profiles of many cells generated from both lineages but also some discrete *Grem1*-lineage clusters. In the interest of clarity, we have not included the integrated scRNAseq dataset shown in **Review Figure 1** below in our revised manuscript, but can include if requested to do so.

Review Figure 1. scRNA transcriptomic analysis of *Grem1*- and *Acan*-lineage articular cells from short-term labelling in the early adult mouse knee. This analysis emphasises mostly shared, but also some discrete, cell clusters generated by these two cartilage lineages. **a**, Experimental schema for isolation of *Grem1*- and *Acan*-lineage articular cells from the early adult mouse knee. **b**, Integration of *Grem1*- and *Acan*-traced datasets was performed using Seurat's 'IntegrateData' function. The anchors for the integration analysis were found using the 'FindIntegrationAnchors' function and canonical correlation analysis (CCA) method, lineage origin from each transgenic mouse line is shown in color (*Grem1*-TdT, orange, *Acan*-TdT cyan). **c**, High *Grem1* (blue), high *Acan* (yellow) or high both *Grem1*/*Acan* expressing (pink) populations are shown.

We further support this observation using co-immunofluorescence staining of knee sections to visualise both discrete *Grem1*-lineage or ACAN+ populations, but also double positive *Grem1*-lineage/ACAN+ cells in the articular cartilage (**Extended Data Fig. 3l**).

This suggests that some *Grem1*-lineage traced cells switch off *Grem1* expression and express *Acan*/ACAN, while others do not. Analysis of differentially expressed transcripts in the *Grem1*-only, compared to *Acan*-only cells, in the *Grem1*-traced scRNAseq dataset identified increased expression of *Is1* which encodes a LIM-domain homeobox transcription factor important for progenitor populations and differentiation across multiple tissues, and decreased expression of multiple collagen transcripts in the *Grem1*-only cells. Together these observations possibly reflect a *Grem1*-expressing progenitor cell population that gives rise to *Acan*/ACAN-expressing cells, although we do not directly address the potential cellular hierarchy of this relationship in this manuscript.

We now also validate functional differences in the *Acan* and *Grem1*-lineages, by comparing the OA phenotype caused by ablation of each cell population using similar genetic models and induction regimens. DT treatment of *Acan*-creERT;*DTR* mice caused a significant decrease in

ACAN-expressing cells, and a mild but significant increase in OA pathology scoring at 26 weeks in comparison to PBS treated controls (**Extended Data Fig. 3o-p**). The overall average OA score in *Acan*-lineage ablated mice was 0.53 (+/- 0.27, st.dev.) in comparison to 1.75 (+/- 0.68) in the *Grem1-TdT-DTR* mice treated with DT (**Extended Data Fig. 3k,o-p**). These data are consistent with previous *Acan*- and *Prg4*- chondrocyte ablation experiments reported to generate mild or insignificant OA (Masson et al., 2019; Zhang et al., 2016).

We have included this additional information in the manuscript, Line 224-271

While we showed that loss of *Grem1*-expressing CP cells is an early event in OA (**Figure 1**) and in turn also causes OA (**Fig. 3, Extended Data Fig. 3**), we considered the possibility that OA may also be caused, in part, by the death or degeneration of the resultant *Grem1*-lineage, mature *Acan*-expressing chondrocytes. To first understand the degree by which *Grem1*-lineage CPs may be distinct from *Acan*-expressing articular chondrocytes, we undertook scRNAseq transcriptomic analysis of FACS-isolated single cells from early adult *Grem1-TdT* knee AC following tamoxifen administration (**Fig. 3m**). This analysis showed that while the majority (72%) of these *Grem1*-lineage traced cells expressed *Acan*, we could also identify *Grem1*-lineage cells that did not express *Acan* (**Fig. 3m**). This was consistent with co-immunofluorescence staining of knee sections in which double positive *Grem1*-lineage/ACAN+ cells were observed in the articular cartilage, but also discrete *Grem1*-lineage and ACAN expressing populations (**Extended Data Fig. 3l**). To investigate potential differences between *Grem1* and *Acan* expressing AC cells, we identified significantly differentially regulated transcripts between *Grem1+Acan*- and *Grem1-Acan*+ populations in our scRNAseq dataset (**Extended Data Fig. 3m**). The most highly upregulated transcript in *Grem1+Acan*- compared to *Grem1-Acan*+ populations was *Isl1* (*Isl1*). *Isl1* encodes an essential LIM-homeodomain transcription factor important for progenitor populations and differentiation across multiple tissues including islet cells and pancreatic mesenchyme²⁶, motor neurons²⁷ and cardiovascular progenitor populations²⁸, but with an underexplored potential function in the AC. Together with differential expression of genes with roles in cartilage or arthritis, such as *platelet derived growth factor receptor alpha* (*Pdgfra*,²⁹), *procollagen C-endopeptidase enhancer 2* (*Pcolce2*,³⁰) and *hyaluronan and proteoglycan link protein 1* (*Hapln1*,³¹), we noted reduced expression of three collagen transcripts (*Col9a2*, *Col9a3*, *Col27a1*) in the *Grem1+Acan*- compared to *Grem1-Acan*+ populations (**Extended Data Fig. 3n**). These transcripts encode key components of the type IX and XXVII extracellular matrix collagens, with vital roles in skeletal and cartilage development in mice and humans³²⁻³⁴. Altogether, this analysis suggested that *Grem1*-expressing articular CPs overlap with the broader *Acan*-expressing population, but are also distinct, and may have distinct functionality generated by differential expression of genes important for cartilage and progenitor populations.

Previous studies noted that ablation of chondrocytes marked by *Acan* or *Prg4* results in mild OA that resolves over time, or no OA^{35,36}. To further discriminate the role of *Acan*+ chondrocytes in comparison to *Grem1*-lineage CPs in OA, we compared the OA phenotype caused by ablation of each cell population using similar genetic models and induction regimens. DT treatment of *Acan-creERT;DTR* mice caused a significant decrease in ACAN-expressing cells, and a mild but significant increase in OA pathology scoring at 26 weeks in comparison to PBS treated controls (**Extended Data Fig. 3o-p**). The overall average OA score in *Acan*-lineage ablated mice was 0.53 (+/- 0.27, st.dev.) in comparison to 1.75 (+/- 0.68) in the *Grem1-TdT-DTR* mice treated with DT (**Extended Data Fig. 3k,o-p**). These data are consistent with previous *Acan*- and *Prg4*- chondrocyte ablation experiments reported to generate mild or insignificant OA^{35,36}. Given these differences, we note that OA is not just a disease of mature chondrocyte loss, but is likely also a failure of local progenitor regeneration.

We next used flow cytometry to isolate *Grem1*-lineage cells from early adult mice and implanted them into recipient mice that had undergone DMM surgery via intra-articular injection. This initial effort to implant *Grem1*-lineage cells was hindered by inefficient homing to and/or survival of the cells in the AC (**Extended Data Fig. 3q**). Thus, we subsequently studied how to preserve this important population through normal aging (**Fig 3**) or injury associated (**Fig. 1**) attrition that could be key for OA prevention.

Discussion line 370-377

Other key differences between the *Grem1*-lineage and that of the broader chondrocytic population marked by *Acan* include the extent and location of labelled lineage cells (**Fig. 2e**, **Extended Data Fig. 2b**), differential expression of cartilage-related transcripts (**Extended Data Fig. 3n**), clonogenicity and differential potential *ex vivo* (**Fig. 2d-i**) and response to injury (**Fig. 1b-f**), such that *Grem1*- but not *Acan*-lineage cells were lost in the early OA-like phenotype induced by DMM surgery. Ablation of *Grem1*-lineage CP cells in the knee joint causes OA, with a milder phenotype observed using similar induction protocols and transgenic mouse models to ablate *Acan*-lineage cells (**Extended Data Fig. 3k,o**).

Reviewer #2 (Remarks to the Author):

The effort put in by the authors to address the Reviewers' comments is appreciated. However, the primary issue of distinguishing *Grem1*⁺ cells from bulk articular chondrocytes remains unaddressed. This would be a very simple experiment that would build on the scRNA-seq data already presented: comparing sorted *Acan*-Cre traced cells with *Grem1*-Cre traced cells. If *Grem1*⁺ cells represent or contain a distinct population of cells, this experiment would clearly demonstrate this and potentially identify regulators of this difference. Given the strength of the claims made by the authors, i.e. that *Grem1*⁺ cells represent chondrocyte progenitors that are also involved in OA pathogenesis, defining these cells as functionally and molecularly distinct is of great importance and fundamental to the paper.

Without consistent use of *Acan*-Cre throughout the paper and comparing the results obtained with *Grem1*-Cre, the claim that *Grem1* marks a chondrogenic progenitor cell is not distinguishable from the claim that *Acan* marks a chondrogenic progenitor cell. The use of *Grem1* as a marker of CPs is the fundamental premise of the paper and the data in their current form do not support this. scRNA-seq would be the simplest way of addressing this, but until some molecular evidence to support the difference between these two populations is presented the paper does not provide much in the way of significance given the known roles of *Foxo1* and *Fgg118* in articular cartilage. In vitro assays and immunostaining are supportive of the claims, but functional (in vivo) and molecular comparisons between *Grem1*- and *Acan*-Cre cells are lacking and sorely needed to support the claims.

We thank the reviewer for this comment and have now included this analysis in the revised manuscript, see point 2 reviewer 1 above.

Reviewer #1 (Remarks to the Author):

The reviewers have adequately addressed the comments raised and this manuscript is seen as suitable for acceptance.